# Unrolled denoising networks provably learn to perform optimal Bayesian inference

**Aayush Karan**[*]
Harvard SEAS
akaran1@g.harvard.edu

**Kulin Shah**[*]
UT Austin
kulinshah@utexas.edu

**Sitan Chen**
Harvard SEAS
sitan@seas.harvard.edu

**Yonina C. Eldar**
Weizmann Institute of Science
yonina.eldar@weizmann.ac.il

## Abstract

Much of Bayesian inference centers around the design of estimators for inverse problems which are optimal assuming the data comes from a known prior. But what do these optimality guarantees mean if the prior is unknown? In recent years, algorithm unrolling has emerged as deep learning's answer to this age-old question: design a neural network whose layers can in principle simulate iterations of inference algorithms and train on data generated by the unknown prior. Despite its empirical success, however, it has remained unclear whether this method can provably recover the performance of its optimal, prior-aware counterparts.

In this work, we prove the first rigorous learning guarantees for neural networks based on unrolling approximate message passing (AMP). For compressed sensing, we prove that when trained on data drawn from a product prior, the layers of the network approximately converge to the same denoisers used in Bayes AMP. We also provide extensive numerical experiments for compressed sensing and rank-one matrix estimation demonstrating the advantages of our unrolled architecture – in addition to being able to obliviously adapt to general priors, it exhibits improvements over Bayes AMP in more general settings of low dimensions, non-Gaussian designs, and non-product priors.

## 1  Introduction

Inverse problems within engineering and the sciences [BBDM21, ST99, Vog02] have inspired the development of a rich toolbox of algorithms for inferring unknown signals given noisy measurements. For instance, a classic approach to solving sparse linear inverse problems is to solve the LASSO using an iterative algorithm like ISTA [DDDM04] or FISTA [BT09]. While these methods are easy to implement and remarkably performant, they are not designed to exploit *distributional* information about the underlying signal, which often comes from domain knowledge. In contrast, Bayesian methods like message passing and variational inference offer a natural framework for designing estimators that incorporate this kind of information: the algorithm designer crafts a *prior* for the signal, and the measurements they observe naturally induce a *posterior* over what the underlying signal could have been.

---

[*]Equal contribution

38th Conference on Neural Information Processing Systems (NeurIPS 2024).

Such an approach comes at a cost. On one hand, this method often comes with strong optimality guarantees in the well-specified setting where the algorithm designer has access to the true prior distribution of the data. In practice, however, this prior is not known *a priori* and hence must be inferred, and any mismatch between the inferred prior and the true distribution will adversely affect the performance of the estimator in ways that are poorly understood compared to what is known in the well-specified setting [BCPS21, BHMS22, MS22].

In recent years, *algorithm unrolling* has emerged as a scalable solution for developing estimators that can improve upon this practicality-performance tradeoff by learning from samples drawn from the data distribution [GL10, MLE21, LTG+20, SWED23]. The idea is to craft a neural network architecture, each of whose layers is expressive enough to implement one step of an existing, hand-crafted iterative algorithm (e.g. ISTA). Then, instead of explicitly setting the weights of the network so that it implements that algorithm, one trains the network on examples of the inference problem at hand using stochastic gradient descent. Remarkably, the algorithm that the network converges to tends to perform at least as well as (and often better than) the hand-crafted algorithm being unrolled, e.g. in the number of layers and iterations necessary to achieve a certain level of error.

Thus, at least empirically, algorithm unrolling seems to achieve the best of both worlds, marrying the domain-aware power of classical iterative methods with the remarkable learning capabilities of neural networks. From a theoretical perspective however, our understanding of its performance is rather limited, as existing guarantees are centered around non-algorithmic aspects like representational power and generalization bounds (see Section 1.1 for a detailed discussion).

In particular, the following fundamental learning question remains open:

*Can an unrolled network trained with stochastic gradient descent provably obtain an estimator competitive with the best prior-aware algorithms?*

In this work, we give the first rigorous learning guarantees for this question, focusing on the well-studied setting of *compressed sensing* (see Section 2.1 for definitions). In addition, we provide the first empirical evidence in the affirmative for the problem of *rank-one matrix estimation* (Section C.1).

**Approximate message passing and unrolling.** Consider the standard Bayesian setup where we observe a noisy measurement $y$ of some signal $x$, and would like to output an estimate $\widehat{x}$ minimizing $\mathbb{E}\|x - \widehat{x}\|^2$. Information-theoretically, the Bayes-optimal estimator for this task is the posterior mean $\mathbb{E}[x \mid y]$, but in many settings of interest this estimator may not be computable by a polynomial-time algorithm. Among computationally efficient estimators, for a wide variety of such inference tasks it is conjectured [MW22b, CMW20, MW22a, BKM+19] that a certain family of iterative algorithms called *approximate message passing (AMP)* is optimal. We give a self-contained exposition of this method in Section 2. Roughly speaking, one can think of AMP as a more advanced version of ISTA where the denoiser at each step can be tuned depending on the prior, and additionally there is a crucial momentum term inspired by a correction from statistical physics [TAP77]. AMP with the optimal tuning of the denoiser is called *Bayes AMP*.

Motivated by the appealing theoretical properties of AMP, in this work we investigate the training dynamics of neural networks given by unrolling this algorithm. In place of the prior-dependent denoisers $\eta_1, \eta_2, \ldots$, we consider generic denoisers given by *neural networks* $\widehat{f}_1, \widehat{f}_2, \ldots$ and unroll the iterations of AMP into layers of a neural network (see Section 2.2 for details). For the theoretical results in this work, we focus on the setting where the only trainable parameters in the network are the ones parametrizing the denoisers $\widehat{f}_\ell$.

Unrolled AMP architectures and variants thereof were originally proposed and empirically investigated by Borgerding et al. [BS16, BSR17] and follow-ups [MMB17, MJC21, ITW19]. These works found that unrolled AMP can significantly outperform unrolled ISTA as well as a version of AMP with soft threshholding denoisers in terms of convergence; i.e., the number of layers needed to achieve a certain MSE.

Despite these compelling experimental results, to our knowledge, there is still little understanding as to whether these networks can actually recover the performance of Bayes AMP. The main theoretical result of this work is to give the first proof that unrolled AMP networks trained with

gradient descent converge to the same denoisers as Bayes AMP and thus achieve mean squared error which is conjectured to be optimal among all polynomial-time algorithms for compressed sensing:

**Theorem 1** (Informal, see Theorem 2). *For compressed sensing with Gaussian sensing matrix, if the prior on the signal is a product distribution with smooth, sub-Gaussian marginals, then an unrolled network based on AMP which is trained with gradient descent on polynomially many samples will converge in polynomially many iterations to an estimator which, in the infinite-dimensional limit, achieves the same mean squared error as Bayes AMP.*

Our proof is based on a novel synthesis of *state evolution*, the fundamental distributional recursion driving analyses of AMP, together with neural tangent kernel (NTK) analysis of training dynamics for overparametrized networks. Crucially, unlike in typical applications of NTK analysis, the level of overparametrization needed in our network is *dimension-independent* even when the second moment $\mathbb{E}_{x \sim q}\|x\|^2$ scales with the dimension $d$. The central reason behind this is that state evolution allows us to map the training dynamics of the network, which *a priori* lives in $L_2(\mathbb{R}^d)$, to training dynamics over the space of functions $L_2(\mathbb{R})$, where the resulting learning problem amounts to that of *one-dimensional score estimation*. As a result, our learning guarantee only requires overparametrization scaling inverse polynomially in the target error.

**Experiments.** We complement these theoretical results with extensive numerical experiments. Our main empirical contributions are as follows:

- We demonstrate that our theoretically motivated unrolled network learns the same optimal denoisers as Bayes AMP, providing a practical alternative that achieves the same performance but does not require explicit knowledge of the true signal prior.
- We observe that introducing auxiliary trainable parameters along with learnable denoisers further improves performance over AMP in low-dimensional settings (where the asymptotic optimality of Bayes AMP does not apply) and when the sensing matrix is non-Gaussian, both in *well-conditioned* and *ill-conditioned* settings.
- We introduce rank-one matrix estimation as a new "model organism" for probing the properties of unrolled networks. To our knowledge, despite its prominence in the theoretical literature on AMP, rank-one matrix estimation has not yet been studied in the context of algorithm unrolling.

The general approach of unrolling with learned denoisers is lesser utilized in the algorithm unrolling literature, which instead largely emphasizes learning auxiliary parameters around domain-specific entities – e.g. measurement matrices or sparse coding dictionaries – while fixing denoisers typically to a soft thresholding function. Our results indicate that learned denoisers can in fact capture distributional priors and are composable with these domain-specific learned parameters, providing a valuable addition to the algorithmic toolkit for practitioners of both AMP and unrolling.

## 1.1 Related work

We provide an extensive review of prior work in the appendix. Here we discuss the works most directly related to ours.

**Theory for unrolling ISTA.** The existing theory for algorithm unrolling almost exclusively focuses on unrolled ISTA (often called LISTA) and compressed sensing. Unlike the present work, they do not consider a Bayesian setting: the signal $x$ is a deterministic sparse vector, and the goal is to converge to $x$. Instead of proving learning guarantees, most of them are representational in nature, arguing that under certain settings of the weights in LISTA, the estimator computed by the network can be more iteration-efficient than vanilla ISTA [MB16, CLWY18, XWG+16, LC19, CLWY21]. The works of [SARE23, SBR23, BRS22] proved generalization bounds for unrolled networks; these are statistical rather than computational in nature. Finally, recent work of [SPP+23] studied optimization aspects of LISTA, and their main theorem, motivated by the NTK literature [LZB20, LZB22] from a different perspective than ours, was an upper bound on the Hessian of the empirical risk of an unrolled ISTA network in a neighborhood around random initialization.

**Unrolled AMP.** Borgerding et al. [BS16, BSR17] were the first to propose unrolling AMP for compressed sensing. In contrast to the architecture we consider, they primarily considered fixed soft-thresholding denoisers $\eta_{st}(\cdot; \lambda)$, in addition to simple parametric families of denoisers like 5-wise linear functions and splines. For these parametric denoisers, on simple priors like Bernoulli-Gaussian they found that the learned network could approach the performance of the "oracle" estimator that knows the support of the underlying signal — see the Appendix for further discussion.

**Other learning-based approaches.** Here we further motivate the setting we consider by contrasting with other possible approaches to learning optimal inference algorithms from data. Perhaps the most obvious would be to simply try to directly learn an approximation to the density function for the prior, e.g. via kernel density estimation or some other non-parametric method. In the product-prior setting in which we prove our results, this is indeed a viable approach in theory. But in practice, unlike algorithm unrolling, this will not scale gracefully to general high-dimensional distributions [GMOV19].

A more scalable approach might training a diffusion model on the data distribution [HD05]. The learned score network could then be used to approximately implement Bayes AMP. Our method is roughly a special case of this: whereas little is known about provable score estimation in general [SCK23, GKL24, CKS24], our theoretical results demonstrate that layerwise training of unrolled networks is a viable, provably correct way to implicitly estimate the score of the data distribution. Furthermore, unrolling accommodates additional trainable parameters to improve robustness to real-world deviations from the stylized models studied in theory.

Finally, we mention the recent theoretical work of [IS24], which shows that semidefinite programs can simulate AMP. While this is not a learning result, it has a similar motivation of reproducing the performance guarantees of AMP using a more robust suite of algorithmic tools.

**Other theory for unrolling.** [MW23] established sample complexity bounds for learning graphical models via diffusion models by unrolling the variational inference algorithms used for score estimation into a ResNet and bounding the number of parameters needed for the network to express these algorithms. Similarly, [Mei24] showed that the popular U-Net architecture can simulate message-passing algorithms. These works can be interpreted as giving representational guarantees for algorithm unrolling, whereas in contrast, the focus of our work is on proving learning guarantees.

## 2 Preliminaries on Bayes AMP and unrolling

Here we give an overview of the Bayes AMP algorithm in the compressed sensing setting. We refer the reader to Appendix C for a full treatment of the rank-one matrix estimation setting. For convenience, when it is clear from context, we use Bayes AMP to refer to the general algorithm used in either setting.

### 2.1 Compressed sensing

In compressed sensing, we are given noisy linear measurements $y \in \mathbb{R}^m$ obtained from an unknown signal $x \in \mathbb{R}^d$ via the observation process

$$y = Ax + \varsigma, \tag{1}$$

where $A \in \mathbb{R}^{m \times d}$ and $\varsigma \in \mathbb{R}^m$ is a random noise vector with i.i.d. entries drawn from the distribution $\mathsf{N}(0, \sigma^2)$. Throughout, we assume that $x \sim p_{\mathsf{x}}$ for product prior $p_{\mathsf{x}} \triangleq p^{\otimes d}$, where $p$ is some distribution over $\mathbb{R}$. The *compressed sensing* problem aims to recover the unknown signal $x$ with estimate $\widehat{x}$ such that the mean squared error (MSE) $\frac{1}{d}\mathbb{E}\|\widehat{x} - x\|^2$ is minimal. Throughout this work we focus on the *proportional asymptotic* setting where we implicitly work with a sequence of such compressed sensing problems indexed by dimension $d$, where $d$ and $m = m(d)$ jointly tend to infinity and $m(d)/d \to \delta$ for absolute constant $\delta > 0$.

[DMM09] proposed the following approximate message passing (AMP) algorithm for estimating $x$ given $A, y$. The algorithm starts with $x_0 = 0$ and $v_0 = y$ and proceeds by

$$x_{\ell+1} = f_\ell(A^\top v_\ell + x_\ell) \tag{2}$$

$$v_\ell = y - Ax_\ell + \frac{1}{\delta} v_{\ell-1} \langle f'_{\ell-1}(A^\top v_{\ell-1} + x_{\ell-1}) \rangle. \tag{3}$$

where $f_\ell : \mathbb{R} \to \mathbb{R}$ is a scalar denoiser applied entrywise, $f'_{\ell-1}$ is also applied entrywise, and $\langle \cdot \rangle$ denotes an entrywise average. The last term in Eq. (3) is commonly referred to as the *Onsager term*. Importantly, the AMP iterates asymptotically satisfy a distributional recursion called *state evolution* [BM11]. Suppose the entries of $A$ are given by $A_{ij} \sim \mathsf{N}(0, 1/m)$. Define the *state evolution parameters* $(\tau_\ell)$ via the scalar recursion

$$\tau_{\ell+1}^2 = \sigma^2 + \frac{1}{\delta}\mathbb{E}\big[(f_\ell(X + \tau_\ell Z) - X)^2\big] \quad \text{with} \quad \tau_0 = \sigma^2 + \frac{1}{\delta}\mathbb{E}[X^2],$$

where $X \sim p$ and $Z \sim \mathsf{N}(0,1)$. Then it is known that as $d \to \infty$, the empirical distribution over entries of $A^\top v_\ell + x_\ell$ converges in a certain sense to the one-dimensional distribution over $X + \tau_\ell Z$ [BM11]. While updates of AMP can be run with any choice of differentiable $f_\ell$, there is an asymptotically optimal choice that depends on the underlying prior $p_{\mathsf{x}}$, and the resulting optimal algorithm is called *Bayes AMP*. In particular, let $\tau_0^{*2} = \sigma^2 + \frac{1}{\delta}\mathbb{E}[X^2]$, and define

$$f_\ell^* = \mathbb{E}[X | X + \tau_\ell^* Z = \cdot] \qquad \text{and} \qquad \tau_{\ell+1}^{*2} = \sigma^2 + \frac{1}{\delta}\mathbb{E}\big[\big(f_\ell^*(X + \tau_\ell^* Z) - X\big)^2\big], \qquad (4)$$

where $X \sim p$ and $Z \sim \mathsf{N}(0,1)$. Then setting $f_\ell = f_\ell^*$ for all $\ell$ in Eqs. (2) and (3) yields Bayes AMP.

In the asymptotic limit, i.e. as $m, d \to \infty$, Bayes AMP has strong theoretical properties. In the setting above, it is conjectured to obtain the optimal MSE over all polynomial-time algorithms [BKM$^+$19] and has been proven to be optimal over a quite general class of algorithms known as *general first-order methods* (*GFOMs*) [CMW20, MW22b].

In practice, however, Bayes AMP is subtly nontrivial to implement. For starters, one must know $p$ to construct $f_\ell^*$. Furthermore, using the exact recursion in Eq. (4) can often lead the algorithm to diverge in finite dimensions. One instead estimates the state evolution parameters from the previous iterates, i.e. replacing $\tau_\ell^2$ with $\frac{1}{m}\|v_\ell\|_2^2$, which is typically enough to stabilize Bayes AMP. The fact that this is a valid estimate follows by state evolution, which ensures that in the infinite dimensional limit, the entries of $v_\ell$ are distributed according to $\mathsf{N}(0, \tau_\ell^2)$ [BM11].

## 2.2 Unrolling Bayes AMP

The aforementioned challenges in realizing the conjectured optimality of Bayes AMP in practice motivate the need for a robust method that does not require knowledge of the prior distribution $p_{\mathsf{x}}$. We consider replacing each scalar denoiser $f_\ell$ in Eq. (2) or (22) with a multilayer perceptron (MLP) that *learns* the "right" denoiser function to use at each iteration of AMP. As we will see, a prudent training approach is enough to provably ensure that our unrolled network learns the optimal denoiser at each layer, effectively recovering Bayes AMP even without explicit knowledge of the prior.

**Architecture.** Suppose we are given training data $\{(y^i, x^i)\}_{i=1}^N$ generated according to Eq. (1) with $x^i \sim p_{\mathsf{x}}$ for all $i$. Let $L$ denote the number of layers in our unrolled network, and let $\mathcal{F}$ denote a family of MLPs with fixed architecture (i.e. fixed depth and width) constrained to a two-dimensional input and one-dimensional output. For each $\ell \in [0, L-1]$, initialize an MLP $\widehat{f_\ell} : \mathbb{R}^2 \to \mathbb{R}$ chosen from $\mathcal{F}$. Set $\widehat{x}_0 = 0_d \in \mathbb{R}^d$ and $\widehat{v}_0 = y^i \in \mathbb{R}^m$, for a given training input $y^i$. Then for each layer $\ell \in [0, L-1]$, our network computes the forward pass

$$\widehat{x}_{\ell+1} = \widehat{f_\ell}(A^\top \widehat{v}_\ell + \widehat{x}_\ell; \widehat{\tau}_\ell) \quad \text{and} \quad \widehat{v}_{\ell+1} = y - A\widehat{x}_{\ell+1} + \frac{1}{\delta}\widehat{v}_\ell \langle \partial_1 \widehat{f_\ell}(A^\top \widehat{v}_\ell + \widehat{x}_\ell; \widehat{\tau}_\ell) \rangle, \quad (5)$$

where $\widehat{\tau}_\ell = \|\widehat{v}_\ell\|_2 / \sqrt{m}$ and $\partial_1$ denotes differentiation with respect to the first input parameter. The notation $\widehat{f_\ell}(\cdot; \widehat{\tau}_\ell)$ denotes applying the scalar function $\widehat{f_\ell}(\cdot, \widehat{\tau}_\ell)$ entrywise. We emphasize that $\widehat{f_\ell}$ is tied to $\partial_1 \widehat{f_\ell}$; that is, we are taking the derivative of the MLP to compute the Onsager term. We refer to our unrolled architecture as an **LDNet** (**L**earned **D**enoising **Net**work).

**Training.** Naïvely, one might consider training the $L$-layer network end-to-end on the mean squared errors of the network estimates – i.e., either with loss function $\mathcal{L}_{CS} = \frac{1}{N}\sum_{i=1}^N \frac{1}{d}\|\widehat{x}_L^i - x^i\|_2^2$ for compressed sensing or $\mathcal{L}_{ME} = \frac{1}{N}\sum_{i=1}^N \frac{1}{d}\|\widehat{x}_L^i \widehat{x}_L^{i\top} - x^i x^{i\top}\|_F^2$ for rank-one matrix estimation.

However, as we observed empirically (echoed by findings in [MMB17]), such an approach gets trapped in suboptimal local assignments of denoising functions.

Instead, we employ *layerwise training*, where we iteratively train the $\ell$-th denoiser $\widehat{f_\ell}$ on the mean squared error loss for the layer-$\ell$ estimate. If $\Psi$ denotes an LDNet with $L$ layers, let $\Psi[0 : \ell]$ denote the subnetwork that consists only of the first $\ell + 1$ layers of $\Psi$, with denoiser $\widehat{f_\ell}$ at layer $\ell$ for $0 \leq \ell \leq L - 1$. Then our training procedure follows Algorithm 1. Note we initialize the $\ell$-th denoiser weights with the previous learned denoiser before training – while this is not relevant to our theoretical results in Section 3, empirically, we find that this initialization is necessary to avoid being trapped in suboptimal regions of parameters. Likewise, we include an optional finetuning step that further reduces approximation error in the learned denoisers but is not needed for our theory results.

---

**Algorithm 1:** Layerwise Training

**Input:** Training data $\mathcal{D}$, LDNet $\Psi$
1 **for** $\ell = 0$ *to* $\ell = L - 1$ **do**
2     **if** $\ell > 0$ **then**
3         $\lfloor$ Initialize $\widehat{f_\ell} \leftarrow \widehat{f_{\ell-1}}$;
4     Freeze learnable weights in $\widehat{f_k}$ for $k < \ell$;
5     Train $\Psi[0 : \ell]$ on $\mathcal{D}$;
    // Optional finetuning step
6     Unfreeze learnable weights in $\widehat{f_k}$ for $k < \ell$ and train $\Psi[0 : \ell]$ on $\mathcal{D}$;

**Output:** Fully trained $\Psi$

---

The proof of optimality of Bayes AMP among all implementations of AMP for the problems we consider, as given in [CMW20, MW22b], strongly motivates our training method: assuming we have learned optimal denoisers up to layer $\ell - 1$, one can show that the minimum mean squared error at layer $\ell$ is achieved by the denoiser used in Bayes AMP. This gives a heuristic sense for how layerwise training facilitates learning optimal denoisers, and this intuition is validated in both our theory and experiments.

## 3 Provably learning Bayes AMP

We now provide theoretical guarantees that our unrolled denoising network can learn Bayes-optimal denoisers when trained in a layerwise fashion. Consider any prior $p_x = p^{\otimes d}$ for which $p$ satisfies the following assumption. The product prior setting is quite standard and widely studied within the theory literature on AMP (e.g. [BM11, DAM16]).

**Assumption 1.** *Given $\tau \geq 0$, let $p(\cdot; \tau)$ denote the density of the convolution $p \star \mathsf{N}(0, \tau^2)$. We assume that:*

*1. $p$ is R-sub-Gaussian with $\mathbb{E}_{X \sim p}[X] = 0$.*

*2. The* score function $\partial_1 p(\cdot; \tau)$ *is B-Lipschitz for all $\tau \geq \sigma^2$, where $\sigma^2$ is the variance of the entries of $\varsigma$ in Eq. (1).*

Both assumptions are relatively mild and hold for a large class of distributions. For example, the sub-Gaussianity holds for any distribution with bounded support (see Section 2.5 in [Ver18]) and the Lipschitzness of the score function is a consequence of regularizing properties of heat flow (see e.g. Lemma 4 in [MS23].

Our main guarantee (see Theorem 2 below) is that under Assumption 1, a suitable unrolled architecture trained with SGD on examples of compressed sensing tasks can compete with Bayes AMP. In Section 3.1, we define the training objective and architecture, describe how our bounds will depend on the underlying prior $p$, and formally state our main result. The full proof is provided in Appendix B.

## 3.1 Proof preliminaries and theorem statement

To prove our learning guarantee, we start by proving that the error in using the learned denoiser in one unrolled layer is small. Denote the sequence generated by the learned denoiser $\widehat{f}$ by $\widehat{x}_\ell$ and $\widehat{v}_\ell$. The Onsager term in AMP ensures that for any iteration $\ell$, the distribution of $A^\top \widehat{v}_\ell + \widehat{x}_\ell$ asymptotically behaves as if every coordinate is i.i.d. as $d \to \infty$ for any $\ell$ (Lemma 1). Therefore, it suffices to learn the denoiser for any fixed coordinate.[2] Without loss of generality we consider the first coordinate and try to learn the denoiser function by minimizing the following objective:

$$\min_g \ \mathbb{E}\big[(g(A_1^\top \widehat{v}_\ell + \widehat{x}_\ell^{(1)}) - x^{(1)})^2\big], \tag{6}$$

where $A_j^\top$ denotes the $j$th row of $A^\top$ and $x^{(j)}$ denotes the $j$th coordinate of the vector $x$. As the learning and generalization guarantee is identical for all $\ell$, we will occasionally drop $\ell$ from the subscript when the context is clear.

**Denoiser complexity.** To quantify the complexity of learning Bayes AMP in terms of the underlying prior $p$, we will work with the following notion of the *complexity* of a class of scalar functions from [AZLL19], which we will apply to the denoisers that arise in Bayes AMP:

**Definition 1** (Scalar function complexity [AZLL19]). *Let $C > 0$ be some sufficiently large absolute constant (e.g. $10^4$). Given any smooth function $\phi : \mathbb{R} \to \mathbb{R}$ and a parameter $\alpha > 0$, we define complexity of $\phi$ at scale $\alpha$ as follows. Suppose $\phi$ admits a power series expansion $\phi(z) = \sum_{i=0}^\infty c_i z^i$. Then*

$$\mathfrak{C}_\varepsilon(\phi, \alpha) \triangleq \sum_{i=0}^\infty \big(1 + (\log(1/\varepsilon)/i)^{i/2}\big) \cdot (C\alpha)^i |c_i| \quad and \quad \mathfrak{C}_s(\phi, \alpha) \triangleq C \sum_{i=0}^\infty (i+1)^{1.75} \alpha^i |c_i|.$$

*Given a class of scalar functions $\mathcal{F}$, we define the* complexity *of $\mathcal{F}$ at scale $\alpha$ by $\mathfrak{C}_\varepsilon(\mathcal{F}, \alpha) = \sup_{\phi \in \mathcal{F}} \mathfrak{C}_\varepsilon(\phi, \alpha)$ (and similarly $\mathfrak{C}_s(\mathcal{F}, \alpha) = \sup_{\phi \in \mathcal{F}} \mathfrak{C}_s(\phi, \alpha)$).*

Intuitively, $\mathfrak{C}_\varepsilon(\mathcal{F}, \alpha)$ and $\mathfrak{C}_s(\mathcal{F}, \alpha)$ both captures how much functions in function class $\mathcal{F}$ can be approximated using a low-degree polynomial. For any function $\phi$ and any $\alpha$, the above complexities are related by $\mathfrak{C}_s(\phi, \alpha) \leq \mathfrak{C}_\varepsilon(\phi, \alpha) \leq \mathfrak{C}_s(\phi, O(\alpha)) \times \mathrm{poly}(1/\varepsilon)$ because $(C\sqrt{\log(1/\varepsilon)}/\sqrt{i})^i \leq e^{O(\log 1/\varepsilon)} = \mathrm{poly}(1/\varepsilon)$ for all $i$. We provide more intuition on how $\mathfrak{C}_\varepsilon(\phi, \alpha)$ and $\mathfrak{C}_s(\phi, \alpha)$ scales with $\varepsilon, \alpha$, under mild assumptions on $\phi$ in Section B.5.

**Main result.** We can now formally state the main theoretical guarantee of this work, namely that layerwise training of LDNet results in performance matching that of Bayes AMP for compressed sensing:

**Theorem 2.** *Suppose the prior distribution $p$ satisfies Assumption 1. Then, for every $\varepsilon_2 \in (0, 1)$ and $\varepsilon_1 \in (0, 1/\mathfrak{C}_s(\mathcal{F}, R(\log 1/\varepsilon_2)^{3/2}))$, there exists*

$$M_0 = \mathrm{poly}(\mathfrak{C}_{\varepsilon_1}(f^*, R(\log 1/\varepsilon_2)^{3/2}), 1/\varepsilon_1) \quad and \quad N_0 = \mathrm{poly}(L\mathfrak{C}_s(f^*, R(\log 1/\varepsilon_2)^{3/2}), 1/\varepsilon_1)$$

*such that the following holds.*

*Let $L$ be any positive integer. Consider an LDNet of depth $L$ with MLP denoisers $\widehat{f}_\ell$ given by the MLP architecture in Eq. (8) with $m \geq M_0$ neurons. Suppose the network is trained by running gradient descent from random initialization with step size $\eta = \tilde{\Theta}(1/(\varepsilon_1 m))$ on $n \geq N_0$ samples of the form $(y^i, x^i)$, where each training example is generated by independently sampling Gaussian matrix $A$ with entries i.i.d. from $\mathsf{N}(0, 1/m)$, sampling $x^i \sim p_x = p^{\otimes d}$, and forming $y^i = Ax^i + \varsigma$ for $\varsigma \sim \mathsf{N}(0, \sigma^2 \cdot \mathrm{Id})$.*

---

[2]In our experiments, we learn the denoiser using all coordinates, but in our theoretical result, we focus on learning using only a single coordinate. The latter is less sample-efficient but more convenient for our proof. It should be possible to prove a guarantee for the later, but it is more cumbersome as we need to prove that the samples obtained by different coordinates are close to being i.i.d. for some notion of closeness, and then generalize the result of [AZLL19] to allow for such samples.

*After $T = \tilde{\Theta}(\mathfrak{C}_s(f_\ell^*, R(\log 1/\varepsilon_2)^{3/2})^2/\varepsilon_1^2)$ steps of gradient descent, with high probability the activations $(\hat{x}_L, \hat{v}_L)$ and denoiser $\hat{f}_L$ at the output layer of the LDNet (see (5)) satisfy*

$$\mathbb{E}_{x,A}\Big[\frac{1}{d}\|\hat{f}_L(A^\top \hat{v}_L + \hat{x}_L; \hat{\tau}_L) - x\|^2\Big] \lesssim \mathrm{MSE}_{\mathrm{AMP}}(L) + \Big(\frac{R^2 B^2}{\delta \sigma^7}\Big)^{L+1}(\varepsilon_1 + \varepsilon_2) + o_d(1),$$

*where $\mathrm{MSE}_{\mathrm{AMP}}(L) \triangleq \mathbb{E}_{x,A}\big[\frac{1}{d}\|f_L^*(A^\top v_L + x_L; \tau_L) - x\|^2\big]$ is the error achieved by running $L$ steps of Bayes AMP.*

Observe that the level of overparametrization in terms of number of samples $n$ and number of hidden neurons $m$ needed in Theorem 2 is *dimension-free*, unlike in typical NTK analyses. This happens because state evolution effectively allows us to convert the learning problem in $d$ dimensions to a learning problem in 1 dimension: we can effectively assume that the entries of $A^\top v_\ell + x_\ell$ converge in an appropriate sense to the distribution of $X + \tau_\ell Z$ for $X \sim p$ and $Z \sim \mathcal{N}(0, 1)$.

This ensures that the learning objective effectively reduces to minimizing $\mathbb{E}[(f_\ell(X + \tau_\ell Z) - X)^2]$ over a parametrized family of denoisers $f_\ell$. The latter objective is often referred to as the *score matching objective* (in one dimension), which is minimized by the Bayes-optimal denoiser $f_\ell^* = \mathbb{E}[X|X + \tau_\ell Z = \cdot]$ at each layer $\ell$. A key component in the proof of Theorem 2 is thus to show that gradient descent can learn this optimal denoiser given one-dimensional training data of the form $(X + \tau_\ell Z, X)$.

As we will show, the runtime for gradient descent is largely dictated by the extent to which these denoisers can be polynomially approximated. *A priori*, one might expect that if degree-$s$ polynomials are needed, then the runtime of the algorithm must scale as $d^{O(s)}$. This would be prohibitively expensive if $s$ is increasing in the dimension $d$. Fortunately however, because we are able to reduce to one-dimensional training dynamics, we ultimately achieve much more favorable scaling in $d$.

## 4 Experiments

We now empirically demonstrate the performance of our proposed architecture and training scheme for unrolling Bayes AMP in a variety of statistical settings. Throughout these experiments, we are motivated by the following questions: **a)** Can our method empirically match the performance of Bayes AMP in settings where the latter is conjectured to be computationally optimal? **b)** In these settings, does our network learn the optimal denoisers? **c)** Are there settings where our methods offer a performance advantage over AMP?

### 4.1 Compressed sensing

**Implementation details.** We set $m = 250, d = 500$ and fix a random Gaussian sensing matrix $A \in \mathbb{R}^{250 \times 500}$. We consider two choices of prior for our experiments: *Bernoulli-Gaussian* and $\mathbb{Z}_2$ (i.e. uniform over $\{1, -1\}^n$). For our unrolled architecture, the family $\mathcal{F}$ of learned MLP denoisers was restricted to three hidden layers, each with 70 neurons and GELU activations. This particular architectural choice was the most convenient for our experiments, but our experimental findings are not particularly sensitive to this. We randomly generated a train and validation dataset $\{y^i, x^i\}_{i=1}^N$ with $N = 2^{15}$ samples by sampling from the prior and using Eq. (1). We train layerwise with finetuning as in Algorithm 1.

For each prior, we also implemented Bayes AMP using the corresponding optimal denoiser. As an additional "semi-prior-aware" baseline, we replace the MLP denoisers in LDNet with "guided denoisers" that have the same functional form as the optimal denoisers but contain trainable parameters; see Appendix E for more details on the precise functional forms used. By convention, we report performance results for all methods by the normalized mean squared error (NMSE) $\|\hat{x} - x\|_2^2/\|x\|_2^2$.

**Bernoulli-Gaussian prior.** Here, each entry of $x$ is independently drawn from a standard normal distribution and set to 0 with probability $1 - \varepsilon$; i.e., $p_x = p^{\otimes d}$ where $p(x) = \varepsilon \mathsf{N}(0, 1; x) + (1 - \varepsilon)\delta(x)$, where $\delta$ denotes the Dirac delta at $x = 0$. To match the setting considered in the

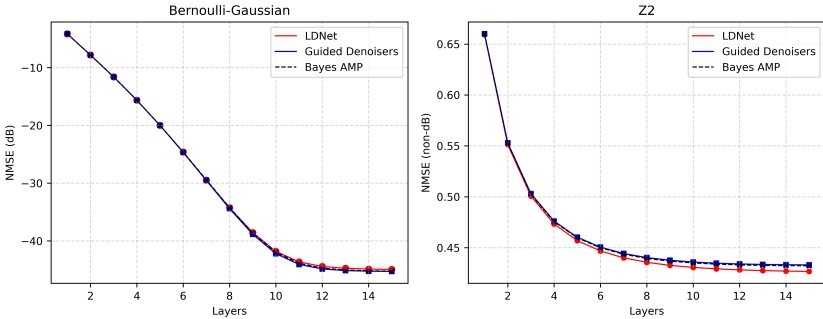

Figure 1: **LDNet for Compressed Sensing**. On the left, we plot the NMSE (in dB) obtained by LDNet and Bayes AMP baselines on the Bernoulli-Gaussian prior. On the right, we plot NMSE (not in dB) achieved on the $\mathbb{Z}_2$ prior. LDNet (along with the guided denoisers) achieves virtually identical performance to the conjectured computationally optimal Bayes AMP.

prior work of Borgerding et al. [BSR17], we set the masking probability to be $\varepsilon = 0.1$ and the measurement noise to be $\sigma^2 = 2 \cdot 10^{-5}$.

We plot the NMSE that our unrolled network and baselines achieve in decibels (dB); that is, $10 \log_{10}(\text{NMSE})$, in Figure 1. LDNet almost perfectly matches the NMSE of Bayes AMP at each layer/iteration. Over 15 layers, our network converges to an NMSE of $-44.9313$ **dB**, as compared to Bayes AMP converging to $-45.3280$ **dB**. As the scale is logarithmic, the difference in error achieved is negligible.

$\mathbb{Z}_2$ **prior.** Here each entry of $x$ is chosen from $\{-1, 1\}$ with probability $\frac{1}{2}$; i.e., $p_x = p^{\otimes d}$ for $p(x) = \frac{1}{2}\delta_{-1}(x) + \frac{1}{2}\delta_{+1}(x)$. To examine a higher noise regime and to ensure Bayes AMP converged within a reasonable number of iterations, we set the measurement noise to be $\sigma^2 = 0.075$. Figure 1 demonstrates that LDNet again recovers Bayes AMP at every iteration, even slightly outperforming by layer 15, achieving an NMSE of 0.4267 (an improvement of 1.28%).

**LDNet denoisers.** From Figure 2 we can observe qualitatively that the learned MLP denoisers recover the functional form for the optimal denoiser at each iteration, as our theory suggests. Interestingly, although the denoisers were trained relative to a fixed sensing matrix, they appear to learn the Bayes AMP denoiser that is measurement-independent, and in Appendix D.3 we show that the performance of these learned denoisers actually transfers to other randomly drawn sensing matrices $A$.

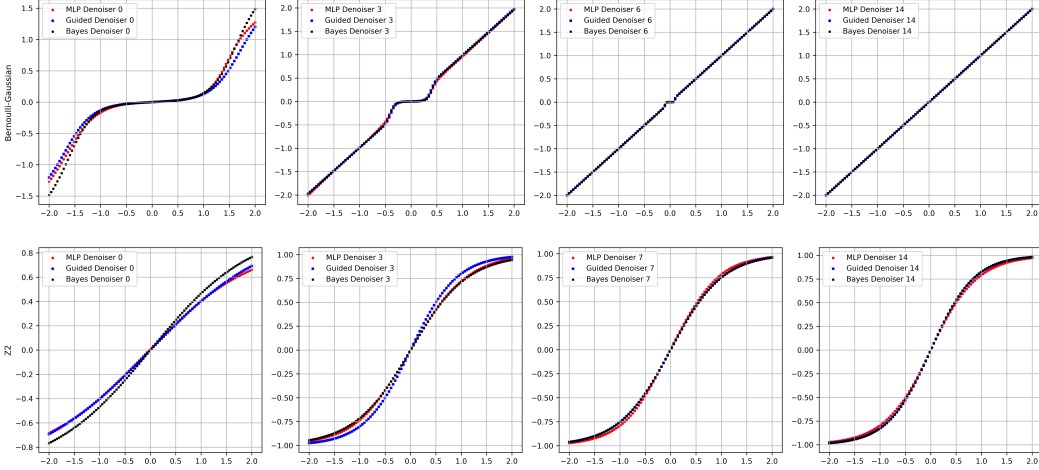

Figure 2: **Learned Denoisers for Compressed Sensing**. We plot layerwise denoising functions learned by LDNet on the Bernoulli-Gaussian and $\mathbb{Z}_2$ priors relative to their optimal denoisers over a range of inputs in $(-2, 2)$. The state evolution input $\tau_\ell$ to each denoiser is set to be its empirical estimate.

## 4.2 Beyond Bayes AMP performance

Much of the algorithm unrolling literature focuses on learning *auxiliary parameters* while using fixed denoisers, as opposed to learning the denoisers themselves. In particular, unrolled methods like LISTA [GL10] and LAMP [BSR17] reparameterize $A^\top$ in Eqs. (2) and (3) as a new learnable matrix $B$, which results in faster convergence than classical, learning-free counterparts like ISTA and AMP.

This does not necessarily contradict Bayes AMP's conjectured optimality for compressed sensing, which only applies in the $d \to \infty$ limit. In the context of our unrolling method, we posit that learning the matrix $B$ can be thought of as learning finite dimensional corrections to the Bayes AMP iterations. In Appendix D.1, we demonstrate that the lower the signal dimensionality, the larger the performance improvement of LDNet with learned matrix $B$ over Bayes AMP.

Finite dimensionality is not the only deviation in design from where Bayes AMP is (conjectured) optimal. In fact we can consider non-Gaussian designs of the measurement matrix $A$, and we show in D.1 that LDNet outperforms Bayes AMP in both well-conditioned and ill-conditioned settings. Furthermore, we can relax the assumption that the signal is drawn for a product prior and extend LDNet to accommodate non-product priors. In D.2, we demonstrate LDNet to surpass Bayes AMP in a non-separable mixture-of-gaussians prior.

# 5 Outlook

In this work we gave the first proof that unrolled denoising networks can compete with optimal prior-aware algorithms simply via gradient-based training on data. Our proof used a novel synthesis of state evolution with NTK theory, and notably, the level of overparametrization needed for our result to hold is independent of the dimension, unlike existing results in the NTK literature. One important consequence of these results is that a one-dimensional score function is learnable with gradient descent, for which only representational, as opposed to algorithm, results existed previously in the literature [MW23].

We supplemented our theory with extensive numerical experiments, confirming that LDNet can recover Bayes AMP performance and Bayes-optimal denoisers without knowledge of the signal prior. Moreover, for various settings where Bayes AMP is not conjectured to perform optimally – e.g. inference in low dimensions, non-Gaussian designs, and non-product priors – we demonstrate that LDNet outperforms Bayes AMP. We thus establish unrolling denoisers as a powerful, practical addition to the algorithmic toolkit for Bayesian inverse problems.

One limitation is that our theoretical results are currently limited to the product prior setting. The non-product setting is difficult because even though state evolution is known here [BMN20], proving an unrolled network converges to the right denoisers essentially amounts to proving that one can learn the score functions of a general data distribution. Additionally, it is subtle to define the right architecture, as the denoisers are no longer scalar, and a generic feedforward architecture would be difficult to prove rigorous guarantees for (and to scale in practice).

In addition, our theoretical results do not immediately extend to the rank-one matrix estimation setting. While closeness in denoising error implies closeness in the state evolution parameter $\tau$ for compressed sensing, this is not immediate for rank-one matrix estimation, where multiple choices for parameters $\mu$ and $\tau$ lead to the same denoising error. This is reflected in Figure 4, where the learned denoisers at early iterations achieve the same MSE as the Bayes optimal denoisers, but the functional forms are completely different. We leave the extension of our compressed sensing results to rank-one matrix estimation as an open question.

Finally, while our experiments suggest that including auxiliary trainable parameters like the "B matrix" offers significant performance advantages once one departs from the asymptotic, Gaussian design setting in which Bayes AMP is believed to be optimal, these are not yet supported by theory. It is an intriguing open question whether one can use some of the insights from the aforementioned representational results for ISTA to rigorously characterize the "non-asymptotic corrections" that these extra learnable parameters are imposing.

## Acknowledgments

AK and SC thank Demba Ba for illuminating discussions about unrolling at an early stage of this project. KS and SC thank Vasilis Kontonis for many fruitful conversations about provable score estimation. AK is supported by the Paul and Daisy Soros Fellowship for New Americans. KS is supported by the NSF AI Institute for Foundations of Machine Learning (IFML).

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

# A  Further related work

**Theory for unrolling ISTA.**   The existing theory for algorithm unrolling almost exclusively focuses on unrolled ISTA (often called LISTA) and compressed sensing. The focus of these works is rather different from ours. For starters, they do not work in a Bayesian setting: the signal $x$ is a deterministic vector assumed to have some level of sparsity, and the goal is to converge to $x$. As mentioned above, these results do not prove convergence guarantees for the *learning algorithm*. Instead, most of them argue that under certain settings of the weights in the LISTA architecture (and variants), the resulting estimator computed by the network can be more iteration-efficient than vanilla ISTA [MB16, CLWY18, XWG$^+$16, LC19, CLWY21].

For example, [MB16] showed that if each layer performs a proximal splitting step with respect to a Gram matrix which admits a factorization with certain nice properties, then the iterates computed by each layer converge to $x$ at an accelerated rate. [CLWY18] showed that if the learned weights are such that the activations converge to the ground truth vector, then they have to have a certain structure; under a specialized LISTA architecture that imposes this structure, they prove there is a setting of weights for which the iterates converge at a linear rate. [XWG$^+$16] showed a similar result for iterative hard thresholding and argued that the learned parameters can potentially reduce the RIP constant of the sensing matrix and thus speed up convergence. [LC19] identified a certain tied parametrization of the weights for LISTA that can also achieve linear convergence with fewer trainable parameters (see also the follow-up [CLWY21]).

Apart from these works, the works of [SARE23, SBR23, BRS22] proved generalization bounds for unrolled networks. These guarantees pertain to questions of sample complexity for empirical risk minimization, instead of computational complexity of learning these networks via gradient descent, and are thus orthogonal to the thrust of our work.

Finally, recent work of [SPP$^+$23] studied optimization aspects of LISTA, and their main theorem was a bound on the Hessian of the empirical risk of an unrolled ISTA network in a neighborhood around random initialization. Under the unproven condition that the associated NTK at initialization is sufficiently well-conditioned, this would imply that the empirical risk satisfies a modified Polyak-Lojasiewicz inequality around initialization and thus the network would converge exponentially quickly to the empirical risk minimizer. While we also draw upon tools from the NTK theory, our focus is not just on optimizing the empirical loss, but on proving that the learned network generalizes to achieve mean squared error competitive with the theoretically optimal prior-aware algorithm, AMP. In addition, our results our end-to-end and apply to unrolled AMP instead of unrolled LISTA.

**Learned AMP.**   Borgerding et al. [BS16, BSR17] were the first to propose unrolling AMP for compressed sensing. In contrast to the architecture we consider, they primarily considered fixed soft-thresholding denoisers $\eta_{st}(\cdot; \lambda)$ with trainable parameter $\lambda$ and trainable weight matrices playing the role of $A^\top$ in the AMP update (see Eqs. (2) and (3)). They found empirically that their unrolled network outperforms AMP with soft-thresholding denoisers. They also considered unrolling vector AMP, a more powerful version of AMP, and showed that even when the sensing matrix is ill-conditioned, the network essentially matches the performance of vector AMP. They also considered some simple parametric families of denoisers like 5-wise linear functions and splines and found that they could approach the performance of the "oracle" estimator that knows the support of the underlying signal.

**Comparison to LDAMP [MMB17].**   Among the various direct follow-ups to [BS16, BSR17], e.g. [ITW19, MJC21, MMB17], the most relevant to ours is the follow-up work of [MMB17] extended this to denoisers given by convolutional neural networks of nontrivial depth (roughly 20 layers). They experimentally demonstrated that these networks performed quite well on compressive image recovery tasks. Using a proof technique of [MMB16] and under a certain monotonicity assumption on the score functions of the data distribution, they showed that Bayes AMP is Bayes optimal (see Lemma 1 therein).

While the fact that they employ a generic architectures for the denoiser step in AMP and demonstrate the effectiveness of the resulting unrolled architecture is closely related to the spirit of the present work, there are important differences. We note that their theoretical result does not not explain how unrolled AMP, when trained on data with gradient descent, learns to

compete with Bayes AMP, only that in certain situations, optimally tuning the denoisers in AMP can achieve Bayes optimality. Furthermore, the monotonicity assumption they make is restrictive (e.g. it does not even apply to the two-point prior given by the uniform distribution over $\{\pm 1\}$). Furthermore, in general Bayes AMP need not be Bayes optimal, i.e. in situations where there is a computational-statistical gap [BPW18].

On the experimental side, our focus was on synthetic setups instead of image recovery, with an emphasis on probing which aspects of the problem setup and which trainable parameters allow unrolled AMP to outperform Bayes AMP.

# B    Proof of main theorem

## B.1    State evolution and learner function

The following result shows that we can characterize the behavior of the AMP iterates $x_\ell$ in the limit as $d \to \infty$.

**Lemma 1** (Asymptotic characterization of AMP iterates [BM11]). *Let $A \in \mathbb{R}^{m \times d}$ be a sensing matrix with i.i.d. entries $A_{ij} \sim \mathsf{N}(0, 1/m)$. Assume $m/d \to \delta \in (0, \infty)$. Consider a sequence of vectors $\{x(d), \eta(d)\}$ indexed by dimension whose empirical distribution converges weakly to probability measures $p_x$ and $p_\eta$ on $\mathbb{R}$ with bounded moments. Then, for any pseudo-Lipschitz function $\psi : \mathbb{R}^t \to \mathbb{R}$ and all $\ell \geq 0$, almost surely*

$$\lim_{d \to \infty} \frac{1}{d} \sum_{i=1}^{d} \psi(x_\ell^{(i)}, x^{(i)}) = \mathbb{E}[\psi(f_\ell(X + \tau_\ell Z), X)]$$

$$\lim_{d \to \infty} \frac{1}{d} \sum_{i=1}^{d} \psi(x^{(i)} - (A_i^\top v_\ell + x_\ell^{(i)}), x^{(i)}) = \mathbb{E}[\psi(\tau_\ell Z, X)].$$

*where $X \sim p_x$ and $Z \sim \mathcal{N}(0, 1)$. The state evolution parameters $\tau_0, \tau_1, \ldots$ are recursively defined as follows*

$$\tau_0^2 = \sigma^2 + \frac{1}{\delta} \mathbb{E}_{X \sim p_x}[X^2]$$

$$\tau_{\ell+1}^2 = \sigma^2 + \frac{1}{\delta} \mathbb{E}_{X \sim p_x, Z \sim \mathcal{N}(0,1)}[(f(X + \tau_\ell Z) - X)^2].$$

(7)

Observe that to minimize the variance $\tau_\ell^2$ at each iteration, the optimal choice of denoiser $f_\ell$ is $f_\ell^*(x) = \mathbb{E}[X \mid X + \tau_\ell Z = x]$.

## B.2    Learning guarantee for a single layer of unrolling

Here we prove a learning result for one layer of unrolled AMP, which we later extend to give an end-to-end learning result for training the full unrolled network.

**Learner function.**    We parametrize the scalar denoiser in a given layer of our unrolled AMP architecture as a one-hidden-layer ReLU network in the following form: letting $w_j^{[t]}$ denote the weight of the $j$th neuron after $t$ steps gradient descent, we consider

$$\widehat{f}(x; \theta_t) = \sum_{j=1}^{m} a_j \mathrm{ReLU}(w_j^{[t]} x + b_j),$$

(8)

We initialize the entries of weights $w_j^{[0]}$ and biases $b_j^{[0]}$ to be i.i.d. from $\mathcal{N}(0, 1/m)$ and entries of $a_j^{[0]}$ to be i.i.d. from $\mathcal{N}(0, \varepsilon_a)$ for some fixed $\varepsilon_a \in (0, 1]$. We only train the weights $w_j$ of hidden layers to simplify the analysis and freeze the bias term $b_j$ and output layer weights $a_j$ at initialization. To update the weights at time $t$, we take one step of online gradient descent

with respect to the loss in Eq. (6) with step size $\eta$ on a fresh sample $(x, y, A)$;[3] here we use the learned denoisers from the previous layers to compute $\widehat{x}_\ell$ and $\widehat{v}_\ell$.

We begin by proving the following claim that training a layer of the unrolled network on gradient descent from random initialization will converge to a denoiser that is competitive with the optimal denoiser under the objective in Eq. (6).

**Lemma 2** (Learning the denoiser within $L_2$ error). *For every $\ell$, assume every coordinate of $\widehat{v}_\ell$ and $\widehat{x}_\ell$ are sub-Gaussian random variables with constant $R$. Then, for every $\varepsilon_2 \in (0, 1)$ and $\varepsilon_1 \in (0, 1/\mathfrak{C}_s(\mathcal{F}, R(\log 1/\varepsilon_2)^{3/2}))$, there exists*

$$M_0 = \text{poly}(\mathfrak{C}_{\varepsilon_1}(f_\ell^*, R(\log 1/\varepsilon_2)^{3/2}), 1/\varepsilon_1) \quad \text{and} \quad N_0 = \text{poly}(\mathfrak{C}_s(f_\ell^*, R(\log 1/\varepsilon_2)^{3/2}), 1/\varepsilon_1)$$

*such that for every $m \geq M_0$ and $n \geq N_0$, choosing learning rate $\eta = \tilde{\Theta}(1/(\varepsilon_1 m))$ and running gradient descent from random initialization for $T = \tilde{\Theta}(\mathfrak{C}_\varepsilon(\phi, R(\log 1/\varepsilon_2)^{3/2})^2 p^2/\varepsilon_1^2)$, with high probability,*

$$\mathbb{E}_{x,y,A}\left[(\widehat{f}_\ell(A_1^\top \widehat{v}_\ell + \widehat{x}_\ell^{(1)}; \theta_T) - x^{(1)})^2\right] \leq \min_{g \in \mathcal{F}} \mathbb{E}\left[(g(A_1^\top \widehat{v}_\ell + \widehat{x}_\ell^{(1)}) - x^{(1)})^2\right] + \varepsilon_1 + \varepsilon_2.$$

This is a consequence of the following result on training neural networks in the NTK regime:

**Lemma 3** (Theorem 1 of [AZLL19][4]). *Consider a target function $F^* : \mathbb{R}^d \to \mathbb{R}$ of the following form*

$$F^*(x) = \sum_{i=1}^{p} a_i^* \phi_i(\langle w_{i,1}^*, x \rangle)\langle w_{i,2}^*, x \rangle$$

*where each $\phi : \mathbb{R} \to \mathbb{R}$ is infinite-order smooth and weights satisfy $\|w_{i,1}^*\|, \|w_{i,2}^*\| \leq 1$ and $|a_i^*| \leq 1$. Additionally, assume that $\|x\| \leq B$. Then, for every $\varepsilon \in (0, 1/(p\mathfrak{C}_\varepsilon(\phi, B)))$, there exists $M_0 = \text{poly}(\mathfrak{C}_\varepsilon(\phi, B), 1/\varepsilon)$ and $N_0 = \text{poly}(\mathfrak{C}_\varepsilon(\phi, B), 1/\varepsilon)$ such that for every $m \geq M_0$ and $n \geq N_0$, choosing learning rate $\eta = \tilde{\Theta}(1/(\varepsilon m))$ and running gradient descent from random initialization for $T = \tilde{\Theta}(\mathfrak{C}_\varepsilon(\phi, B)^2 p^2/\varepsilon^2)$, with high probability,*

$$\mathbb{E}[(N(x; \theta_T) - y)^2] \leq \inf_{g \in \mathcal{F}} \mathbb{E}[(g(x) - y)^2] + \varepsilon.$$

*Additionally, the absolute value of the neural network is bounded by $|N(x; \theta_t)| \lesssim \tilde{\Theta}(\mathfrak{C}_\varepsilon(\phi, B))$ for all $x$ with $\|x\| \leq B$ for all $t$.*

*Proof of Lemma 2.* As the distribution over $\widehat{v}_\ell$ is $R$-sub-Gaussian, for a fixed $A$, we have $|A_i^\top \widehat{v}_\ell| \leq R\|A_i\|\sqrt{\log(1/\varepsilon_2)}$ with probability at least $1 - \varepsilon_2$. Additionally, because of $A_{ij} \sim \mathcal{N}(0, \text{Id}/m)$, we have $\|A_i\| \lesssim \log(1/\varepsilon_2)$ with probability at least $1 - \varepsilon_2$. Combining both bounds, we have $|A_i^\top \widehat{v}_\ell| \lesssim R(\log(1/\varepsilon_2))^{3/2}$. Similarly, using sub-Gaussianity of $\widehat{x}_\ell$, we have $|\widehat{x}_\ell^{(i)}| \lesssim R\sqrt{\log(1/\varepsilon_2)}$. This gives us that $|A_i^\top \widehat{v}_\ell + \widehat{x}_\ell^{(i)}| \lesssim R(\log(1/\varepsilon_2))^{3/2}$ with probability at least $1 - \varepsilon_2$.

By only considering samples satisfying $|A_i^\top \widehat{v}_\ell + \widehat{x}_\ell^{(i)}| \lesssim R(\log(1/\varepsilon_2))^{3/2}$,[5] we can apply Lemma 3 to obtain a neural network that achieves $\varepsilon_1$ error. □

---

[3]In our experiments, we keep the measurement matrix $A$ fixed and only sample fresh $(x, y)$ pairs, but in our theoretical result, we assume that gradient descent uses fresh samples $(x, y, A)$ to avoid technical difficulties regarding dependencies between the errors at different layers of unrolled architecture. We expect that with more work, one can extend the proof to fixed $A$.

[4]Here we state the result in terms of gradient descent instead of *stochastic* gradient descent as in [AZLL19] However, the same proof of [AZLL19] goes through upon slightly modifying Lemma B.4 therein. In Lemma B.4, we can write $\|W_{t+1} - W^*\|_F^2 = \|W_t - \eta\nabla L_F(\mathcal{Z}, W_t) - W^*\|_F^2$ and therefore, getting the equality of $2\eta\langle W_t - W^*, \nabla L_F(\mathcal{Z}, W_t)\rangle = (\|W_t - W^*\|_F^2 - \|W_{t+1} - W^*\|_F^2)/2\eta + (\eta/2)\|\nabla L_F(\mathcal{Z}, W_t)\|_F^2$ and using this equality in Eq.(B.7) of [AZLL19]. The rest of the proof remains the same.

[5]The reason we can do this is that this condition fails to hold only with probability at most $\varepsilon_2$, and whenever it fails to hold, we can output 0 and pay an additional $\varepsilon_2 \cdot \mathbb{E}[x^2]$. Alternatively, we could also modify the learner network to implement the indicator of $|A_i^\top \widehat{v}_\ell + \widehat{x}_\ell i| \lesssim R(\log(1/\varepsilon_2))^{3/2}$ using an appropriate linear combination of ReLU activations without learnable parameters.

Next, we relate the error in Lemma 2, which is for predicting the first coordinate of the signal given the noisy estimate provided by the previous layer of the unrolled network, to the optimal error for predicting a sample from the univariate prior $p$ given a Gaussian corruption. This will follow by state evolution.

**Lemma 4.** *Let $\widehat{f}_\ell(\cdot, \theta_t)$ be the learned neural network using gradient descent after $t$ timesteps such that the conditions of Lemma 2 satisfies. Then, with high probability, we have*

$$\lim_{d\to\infty} \mathbb{E}_{x,A}\big[\frac{1}{d}\|\widehat{f}_\ell(A^\top \widehat{v}_\ell + \widehat{x}_\ell, \theta_T) - x\|^2\big] \leq \mathbb{E}\big[(\widehat{f}_\ell^*(X + \widehat{\tau}_\ell Z) - X)^2\big] + \varepsilon_1 + \varepsilon_2 \,.$$

*Additionally, the following statement holds with high probability:*

$$\mathbb{E}\big[(\widehat{f}_\ell(X + \widehat{\tau}_\ell Z, \theta_T) - X)^2\big] \leq \mathbb{E}\big[(\widehat{f}_\ell^*(X + \widehat{\tau}_\ell Z) - X)^2\big] + \varepsilon_1 + \varepsilon_2 \,.$$

*Proof.* Observe that $A_j^\top \widehat{v}_\ell + \widehat{x}_\ell^{(j)}$ follows the same distribution for all $j$. Therefore, using Lemma 2, we obtain that

$$\mathbb{E}_{x,A}\big[(\widehat{f}_\ell(A_j^\top \widehat{v}_\ell + \widehat{x}_\ell^{(j)}, \theta_T) - x^{(j)})^2\big] \leq \min_g \mathbb{E}\big[(g(A_j^\top \widehat{v}_\ell + \widehat{x}_\ell^{(j)}) - x^{(j)})^2\big] + \varepsilon_1 + \varepsilon_2.$$

As the distribution of $A_j^\top \widehat{v}_\ell + \widehat{x}_\ell^{(j)}$ is the same for all $j \in [d]$, we use the same learned denoiser for all the coordinates. Therefore, we can rewrite the above equation as

$$\frac{1}{d}\sum_{i=1}^T \mathbb{E}_{x,A}\big[\|\widehat{f}(A^\top \widehat{v}_\ell + \widehat{x}_\ell, \theta_T) - x\|^2\big] \leq \min_g \frac{1}{d}\mathbb{E}\big[\|g(A^\top \widehat{v}_\ell + \widehat{x}_\ell) - x\|^2\big] + \varepsilon_1 + \varepsilon_2.$$

Now, we want to use state evolution as $d \to \infty$. Note that as $d \to \infty$, using Lemma 1 with $\psi$ function as $\psi(a, b) = (a - b)^2$, we have

$$\lim_{d\to\infty} \frac{1}{d}\|g(A^\top \widehat{v}_\ell + \widehat{x}_\ell) - x\|^2 = \mathbb{E}\big[(g(X + \widehat{\tau}_\ell Z) - X)^2\big]$$

for any function $g \in \mathcal{F}$. As the quantity inside expectation converges to $\mathbb{E}\big[(g(X + \widehat{\tau}_\ell Z) - X)^2\big]$, using monotone convergence theorem, we have

$$\lim_{d\to\infty} \min_g \mathbb{E}[\frac{1}{d}\|g(A^\top \widehat{v}_\ell + \widehat{x}_\ell) - x\|^2] = \min_g \lim_{d\to\infty} \mathbb{E}[\frac{1}{d}\|g(A^\top \widehat{v}_\ell + \widehat{x}_\ell) - x\|^2]$$

$$= \min_g \mathbb{E}\big[(g(X + \widehat{\tau}_\ell Z) - X)^2\big].$$

Using this result and applying limits on both sides of Lemma 2, we obtain

$$\lim_{d\to\infty} \mathbb{E}_{x,A}\big[\frac{1}{d}\|\widehat{f}_\ell(A^\top \widehat{v}_\ell + \widehat{x}_\ell, \theta_T) - x\|^2\big] \leq \min_g \mathbb{E}\big[(g(X + \widehat{\tau}_\ell Z) - X)^2\big] + \varepsilon_1 + \varepsilon_2.$$

The minimizer of $\mathbb{E}\big[(g(X + \widehat{\tau}_\ell Z) - X)^2\big]$ is given by $\widehat{f}_\ell^*(\cdot) = \mathbb{E}[X \mid X + \widehat{\tau}_\ell Z = \cdot]$. Using this fact, we obtain the first result of the lemma statement. Similar to the proof of the right side of the inequality, the quantity on the left side converges to $\mathbb{E}\big[(\widehat{f}_\ell(X + \widehat{\tau}_\ell Z, \theta_T) - X)^2\big]$ using the monotone convergence theorem. This gives the second result of the lemma statement. $\qquad\square$

### B.3   Stability of optimal denoisers

The right-hand side of the bound in the above Lemma corresponds to the minimum mean squared error achievable for denoising at noise scale $\widehat{\tau}_\ell$, where $\widehat{\tau}_\ell$ is the state evolution parameter corresponding to running the *learned* AMP iterations up to that layer of the unrolled network. To show that the learned network can compete with Bayes AMP, we need to relate $\widehat{\tau}_\ell$ to the corresponding state evolution parameter $\tau_\ell$ given by Bayes AMP. For this, we need the following stability result showing that the minimum mean-squared error is stable with respect to perturbations of the noise scale.

**Lemma 5.** *Let prior $X$ be such that the score function $\partial_1 p(\cdot; \tau)$ is B-Lipschitz continuous for all $\tau$ where $p(\cdot; \tau)$ denotes the probability density function of random variable $X + \tau Z$. Additionally, assume that the variance of $X$ is bounded by $V$. Then, we have*

$$\mathbb{E}[(\widehat{f}_\ell^*(X + \widehat{\tau}_\ell Z) - X)^2] \lesssim \mathbb{E}[(f_\ell^*(X + \tau_\ell Z) - X)^2] + \frac{V^2 B^2}{\sigma^6}|\widehat{\tau}_\ell - \tau_\ell|$$

*where $\widehat{f}_\ell^*(x) = \mathbb{E}[X | X + \widehat{\tau}_\ell Z = x]$ and $f_\ell^*(x) = \mathbb{E}[X | X + \tau_\ell Z = x]$.*

*Proof.* Rewriting the error between $f_\ell^*$ and $\widehat{f}_\ell^*$, we have

$$\mathbb{E}[(\widehat{f}_\ell^*(X + \widehat{\tau}_\ell Z) - X)^2] = \mathbb{E}[(\widehat{f}_\ell^*(X + \widehat{\tau}_\ell Z) - \widehat{f}_\ell^*(X + \tau_\ell Z) + \widehat{f}_\ell^*(X + \tau_\ell Z) - X)^2] \tag{9}$$

$$= \mathbb{E}[(\widehat{f}_\ell^*(X + \widehat{\tau}_\ell Z) - \widehat{f}_\ell^*(X + \tau_\ell Z))^2] \tag{10}$$

$$+ 2\mathbb{E}[(\widehat{f}_\ell^*(X + \widehat{\tau}_\ell Z) - \widehat{f}_\ell^*(X + \tau_\ell Z))(\widehat{f}_\ell^*(X + \tau_\ell Z) - X)] \tag{11}$$

$$+ \mathbb{E}[(\widehat{f}_\ell^*(X + \tau_\ell Z) - X)^2]. \tag{12}$$

The term in Eq. (10) can be upper bounded by $(B(\widehat{\tau}_\ell - \tau_\ell))^2 / \widehat{\tau}_\ell^4$ because $\widehat{f}_\ell$ is Lipschitz by assumption. Using Cauchy-Schwartz for the term in Eq. (11), the second term is upper bounded by

$$2\mathbb{E}[(\widehat{f}_\ell^*(X + \widehat{\tau}_\ell Z) - \widehat{f}_\ell^*(X + \tau_\ell Z))(\widehat{f}_\ell^*(X + \tau_\ell Z) - X)]$$

$$\leq 2\mathbb{E}[(\widehat{f}_\ell^*(X + \widehat{\tau}_\ell Z) - \widehat{f}_\ell^*(X + \tau_\ell Z))^2]^{1/2}\mathbb{E}[(\widehat{f}_\ell^*(X + \tau_\ell Z) - X)^2]^{1/2}$$

$$\leq 2BV|\widehat{\tau}_\ell - \tau_\ell|/\widehat{\tau}_\ell^2. \tag{13}$$

The squared term in Eq. (12) can be upper bounded by

$$\mathbb{E}[(\widehat{f}_\ell^*(X + \tau_\ell Z) - X)^2] = \mathbb{E}[((\widehat{f}_\ell^*(X + \tau_\ell Z) - f_\ell^*(X + \tau_\ell Z))^2)]$$

$$+ 2\mathbb{E}[(\widehat{f}_\ell^*(X + \tau_\ell Z) - f_\ell^*(X + \tau_\ell Z))(f_\ell^*(X + \tau_\ell Z) - X)]$$

$$+ \mathbb{E}[(f_\ell^*(X + \tau_\ell Z) - X)^2].$$

We can upper bound the first term above using Lemma 6. Likewise, the second term can be bounded by Cauchy-Schwarz and Lemma 6. In this way, using that $V^2 \geq \widehat{\tau}_\ell \geq \tau_\ell \geq \sigma^2$, we obtain that

$$\mathbb{E}[(\widehat{f}_\ell^*(X + \tau_\ell Z) - X)^2] \lesssim \mathbb{E}[(f_\ell^*(X + \tau_\ell Z) - X)^2] + \frac{V^2 B^2}{\sigma^6}(\widehat{\tau}_\ell - \tau_\ell)^2 + \frac{V^2 B}{\sigma^3}|\widehat{\tau}_\ell - \tau_\ell|.$$

Combining this bound with Eq. (13), we obtain the bound. $\qquad\square$

The proof above relies on the following "score perturbation lemma" showing that the optimal denoiser is Lipschitz with respect to the noise scale.

**Lemma 6** (Score perturbation lemma)**.** *Let $p(\cdot; \tau)$ be the density function of $X + \tau Z$. Let the score function $\partial_1 \log p(\cdot; \tau)$ be $B$−Lipschitz continuous. Then, the $L_2$ error between the optimal denoisers at two noise scales $\tau_\ell$ and $\widehat{\tau}_\ell$ is given by*

$$\mathbb{E}\left[(\widehat{f}_\ell^*(X + \tau_\ell Z) - f_\ell^*(X + \tau_\ell Z))^2\right] \leq \widehat{\tau}_\ell^2 B^2 (\widehat{\tau}_\ell - \tau_\ell)^2 + \frac{B^2(\widehat{\tau}_\ell - \tau_\ell)^4}{\tau_\ell^4 \widehat{\tau}_\ell^2} V^2.$$

*Proof.* Denote the probability density function of $X + \tau_\ell Z$ random variable at value $x$ as $p(x; \tau_\ell)$. Assuming $\partial_1 \log p(\cdot; \tau_\ell)$ is $B$-Lipschitz function and using Lemma C.11 of [LLT22], we have

$$\mathbb{E}_{x \sim p(\cdot; \tau_\ell)}\left[(\partial_1 \log p(x; \tau_\ell) - \partial_1 \log p(x; \widehat{\tau}_\ell))^2\right] \lesssim B^2(\widehat{\tau}_\ell - \tau_\ell)^2 + B^2(\widehat{\tau}_\ell - \tau_\ell)^4 \mathbb{E}\left[(\partial_1 \log p(x; \tau_\ell))^2\right].$$

Using Tweedie's formula ($\partial_1 \log p(x; \tau_\ell) = (\mathbb{E}[X | X + \tau_\ell Z = x] - x)/\tau_\ell^2$), we have

$$\mathbb{E}_{x \sim p(\cdot; \tau_\ell)}\left[(\partial_1 \log p(x; \tau_\ell) - \partial_1 \log p(x; \widehat{\tau}_\ell))^2\right]$$

$$= \mathbb{E}_{x \sim p(x; \tau)}\left[\left(\frac{(\widehat{\tau}_\ell^2 - \tau_\ell^2)(\mathbb{E}[X | X + \tau_\ell Z = x] - x) + \tau_\ell^2(\mathbb{E}[X | X + \tau_\ell Z = x] - \mathbb{E}[X | X + \widehat{\tau}_\ell Z = x])}{\tau_\ell^2 \widehat{\tau}_\ell^2}\right)^2\right].$$

This implies that

$$\mathbb{E}\big[(\mathbb{E}[X|X+\tau_\ell Z = x] - \mathbb{E}[X|X+\widehat{\tau}_\ell Z = x])^2\big]$$

$$\lesssim \widehat{\tau}_\ell^2 \mathbb{E}_{x\sim p(\cdot;\tau_\ell)}\big[(\partial_1 \log p(x;\tau_\ell) - \partial_1 \log p(x;\widehat{\tau}_\ell))^2\big] + \frac{(\widehat{\tau}_\ell^2 - \tau_\ell^2)^2}{\tau_\ell^4 \widehat{\tau}_\ell^4} \mathbb{E}\big[(\mathbb{E}[X|X+\tau_\ell Z = x] - x)^2\big]$$

$$\lesssim \widehat{\tau}_\ell^2 B^2 (\widehat{\tau}_\ell - \tau_\ell)^2 + \frac{B^2(\widehat{\tau}_\ell - \tau_\ell)^4}{\tau_\ell^4 \widehat{\tau}_\ell^2} \mathbb{E}\big[(\mathbb{E}[X|X+\tau_\ell Z = x] - x)^2\big]$$

where the last inequality uses the fact that $\widehat{\tau}_\ell \geq \tau_\ell$ because $\tau_\ell$ is obtained using optimal denoiser. Additionally, we have $\tau_0 = \sigma^2 + (1/\delta)\mathbb{E}[X^2] \leq \tau^2 + V^2$. Using law of total variance, we have $\text{var}[X] = \mathbb{E}[\text{var}(X|X+\tau_\ell Z)] + \text{var}(\mathbb{E}[X|X+\tau_\ell Z])$. Therefore, we have $\mathbb{E}[X^2] \geq \mathbb{E}[(\mathbb{E}[X|X+\tau_\ell Z = x] - x)^2]$ which implies the claimed bound. $\qquad\square$

## B.4 Putting everything together

We are now ready to conclude the proof of our main result.

*Proof of Theorem 2.* Recall that the state evolution recursion for $\tau_\ell$ is defined using the optimal denoiser $f_\ell^*$ at layer $\ell$ in Lemma 1. Similarly, the state evolution recursion for $\widehat{\tau}_\ell$ is applied using learned denoiser $\widehat{f}_\ell(\cdot, \theta_T)$. At some places, we will use $\widehat{f}_\ell(\cdot)$ to denote $\widehat{f}_\ell(\cdot, \theta_T)$ for brevity. By the state evolution recursion for $\tau_\ell$, the error between $\tau_\ell$ and $\widehat{\tau}_\ell$ is given

$$\widehat{\tau}_\ell^2 - \tau_\ell^2 = \frac{1}{\delta}(\mathbb{E}[(\widehat{f}_{\ell-1}(X+\widehat{\tau}_{\ell-1}Z) - X)^2] - \mathbb{E}[f_{\ell-1}^*(X+\tau_{\ell-1}Z) - X)^2])$$

$$\lesssim \frac{1}{\delta}\Big(\frac{V^2B^2}{\sigma^6}|\widehat{\tau}_{\ell-1} - \tau_{\ell-1}|^2 + \varepsilon_1 + \varepsilon_2\Big),$$

where the last inequality follows from Lemma 4 and Lemma 5. As $\widehat{\tau}_\ell + \tau_\ell \geq 2\sigma$ for all $\ell$ and $\widehat{\tau}_\ell \geq \tau_\ell$, we have

$$|\widehat{\tau}_\ell - \tau_\ell| \lesssim \frac{V^2B^2}{\delta\sigma^7}|\widehat{\tau}_{\ell-1} - \tau_{\ell-1}| + \frac{\varepsilon_1 + \varepsilon_2}{\delta\sigma}.$$

Solving this recurrence, we have that the error in the state evaluation parameters after $L$ layers is upper bounded by

$$|\widehat{\tau}_L - \tau_L| \lesssim \Big(\frac{V^2B^2}{\delta\sigma^7}\Big)^L (\varepsilon_1 + \varepsilon_2).$$

Combining this bound with Lemma 4 and Lemma 5, we have

$$\lim_{d\to\infty} \mathbb{E}_{x,A}\Big[\frac{1}{d}\|\widehat{f}_\ell(A^\top \widehat{v}_\ell + \widehat{x}_\ell, \theta_T) - x\|^2\Big] \lesssim \mathbb{E}[(f_\ell^*(X+\tau_\ell Z) - X)^2] + \Big(\frac{V^2B^2}{\delta\sigma^7}\Big)^{L+1} (\varepsilon_1 + \varepsilon_2).$$

Applying state evolution (Lemma 1), we get that

$$\lim_{d\to\infty} \mathbb{E}_{x,A}\Big[\frac{1}{d}\|\widehat{f}_\ell(A^\top \widehat{v}_\ell + \widehat{x}_\ell, \theta_t) - x\|^2\Big]$$

$$\lesssim \lim_{d\to\infty} \mathbb{E}_{x,A}\Big[\frac{1}{d}\|f_\ell^*(A^\top v_\ell + x_\ell, \theta_t) - x\|^2\Big] + \Big(\frac{V^2B^2}{\delta\sigma^7}\Big)^{L+1} (\varepsilon_1 + \varepsilon_2)$$

as claimed. $\qquad\square$

## B.5 Bounding the complexity of denoisers

In Definition 1, we primarily focus on the setting where $C\alpha \geq 1$. Here we provide some bounds on $\mathfrak{C}_\varepsilon(\phi, \alpha)$ in this regime. This in turn provides the bound for $\mathfrak{C}_s(\phi, \alpha)$. First note that we always have

$$(\log(1/\varepsilon)/i)^{i/2}(C\alpha)^i \leq \exp(\log(1/\varepsilon)C\alpha/2) = (1/\varepsilon)^{O(C\alpha)}. \tag{14}$$

First consider the case where $\phi = \sum_i c_i z^i$ is a degree-$k$ polynomial. Eq. (14) then implies

$$\mathfrak{C}_\varepsilon(\phi, \alpha) \lesssim k(1/\varepsilon)^{O(C\alpha)} \max_i |c_i|. \tag{15}$$

This bound is useful when the degree is high, e.g. when $k = \tilde{\omega}(C\alpha \log(1/\varepsilon))$. We also have the following naive bounds when the degree is small or intermediate. When $k \geq \log(1/\varepsilon)$,

$$\mathfrak{C}_\varepsilon(\phi, \alpha) \lesssim (\log(1/\varepsilon)C\alpha)^{O(\log 1/\varepsilon)} \max_{i \leq \log(1/\varepsilon)} |c_i| + (C\alpha)^k \max_{i > \log(1/\varepsilon)} |c_i| \,. \tag{16}$$

When $k \leq \log(1/\varepsilon)$,

$$\mathfrak{C}_\varepsilon(\phi, \alpha) \lesssim (\log(1/\varepsilon)C\alpha)^{O(k)} \max_i |c_i| \,. \tag{17}$$

In the context of our learning result, it suffices for the true denoisers $f_\ell^*$ to be *well-approximated* by functions of low complexity, e.g. low-degree polynomials.

In our setting of Lipschitz denoisers from Assumption 1, we can get good approximation by low-degree polynomials via the following standard result:

**Lemma 7** (Jackson's theorem [Jac11]). *Let $k \in \mathbb{Z}$, $B, R \geq 0$. If a function $f : \mathbb{R} \to \mathbb{R}$ is B-Lipschitz, then there exists a polynomial $q$ of degree-$k$ for which*

$$\sup_{|z| \leq R} |f(z) - q(z)| \lesssim BR/k \,. \tag{18}$$

As the true denoisers are Lipschitz and the prior $p$ is assumed to be $R$-sub-Gaussian so that the effective support over which the true denoisers must be approximated has radius $O(R)$, this implies that we can approximate them pointwise to error $\delta$ with polynomials of degree $k = O(R/\delta)$. To apply any of the complexity bounds in Eqs. (15)- (17) to these polynomials, it remains to bound the coefficient of largest magnitude. For this, we can apply a result like the following:

**Lemma 8** (Corollary of Lemma 4.1 from [She12]). *Given a polynomial $q(z) = \sum_{i=0}^k c_i z^i$ for which $\sup_{z \in [0,1]} |q(z)| \leq M$,*

$$\sum_i |c_i| \leq 4^d M \,. \tag{19}$$

In situations where the denoiser has additional smoothness properties, one can obtain even better polynomial approximations. To illustrate this, we provide an example in the special case of the $\mathbb{Z}_2$ prior:

**Example 1.** *When $p = \mathbb{Z}_2$, the optimal denoisers used in Bayes AMP are of the form $\tanh(\cdot/\tau^2)$ for various $\tau \geq \sigma$, where $\sigma^2$ is the variance of the measurement noise. This function is $B = 1/\sigma^2$-Lipschitz, and the effective support of $p \star \mathsf{N}(0, \tau^2)$ is of radius $R = O(1)$. By standard results, $\tanh(\cdot/\tau^2)$ can be $\eta$-approximated over $[-O(1), O(1)]$ with a polynomial of degree-$k = O(\log(\sigma/\eta)/\sigma^2)$, see e.g. Lemma 2 in [LSSS14]. Applying Lemma 8 to bound the coefficients of this polynomial by $(\sigma/\eta)^{O(1/\sigma^2)}$ and then applying the bound in Eq. (16), we conclude that the denoisers in the case of $\mathbb{Z}_2$ prior are $\eta$-approximated by polynomials $q$ of complexity*

$$\mathfrak{C}_\varepsilon(q, \alpha) \lesssim (\sigma/\eta)^{O(1/\sigma^2)} \cdot \left( (\log(1/\varepsilon)C\alpha)^{O(\log 1/\varepsilon)} + (C\alpha)^{O(\log(\sigma/\eta)/\sigma^2)} \right) \,. \tag{20}$$

## C Rank-one matrix estimation

### C.1 Preliminaries

While compressed sensing involves a *linear* noisy transformation of its underlying signal, rank-one matrix estimation offers a *nonlinear* counterpart. Here, the unknown signal $x \in \mathbb{R}^d$ is first transformed into a rank-one matrix $xx^\top$. The observed signal is given by

$$Y = \frac{\lambda}{d} x x^T + G, \tag{21}$$

where $G \in \mathbb{R}^{d \times d}$ is a symmetric matrix with entries $G_{ij} \sim \mathsf{N}(0, \frac{1}{d})$ for $i \leq j$ and $\lambda > 0$ denotes the *signal-to-noise* (SNR) ratio. Again, we assume $x \sim p_{\mathsf{x}}$ for product prior $p_{\mathsf{x}} \triangleq p^{\otimes d}$ with $p$ some distribution over $\mathbb{R}$. Then, the *rank-one matrix estimation* problem attempts to recover an

estimate $\widehat{x}$ for $x$ such that the *Frobenius mean squared error* (MSE) $\frac{1}{d}\mathbb{E}\|\widehat{x}\widehat{x}^\top - xx^\top\|_F^2$ is minimal. The asymptotic setting corresponds to working with a sequence of such problems indexed by dimension $d$, where $d \to \infty$.

The approximate message passing (AMP) algorithm for estimating $x$ given $Y$ proposed in [RF12] takes the form

$$x_{\ell+1} = f_\ell(v_\ell) \tag{22}$$

$$v_\ell = Yx_\ell - x_{\ell-1}\langle f'_{\ell-1}(v_{\ell-1})\rangle, \tag{23}$$

where $f_\ell : \mathbb{R} \to \mathbb{R}$ again indicates a scalar denoiser applied componentwise. There are various possible choices for initialization; here we consider $x_0 = \widehat{1} \in \mathbb{R}^d$ and $v_0 = Yx_0$.

As with compressed sensing, AMP iterates for rank-one matrix estimation satisfy a state evolution recursion. Define parameters $\mu_\ell$ and $\tau_\ell$ evolving according to the scalar equations

$$\mu_{\ell+1} = \lambda\mathbb{E}[Xf_\ell(\mu_\ell X + \sqrt{\tau_\ell}Z)] \tag{24}$$

$$\tau_{\ell+1} = \mathbb{E}[f_\ell(\mu_\ell X + \sqrt{\tau_\ell}Z)^2], \tag{25}$$

where $X \sim p$ and $Z \sim \mathsf{N}(0,1)$. Then for $d \to \infty$, the empirical distribution over entries of $v_\ell$ converges asymptotically to the one-dimensional distribution over $\mu_\ell X + \sqrt{\tau_\ell}Z$ [RF12]. Bayes AMP thus corresponds to the choice of denoising functions optimizing the posterior mean on $X$ given $\mu_\ell X + \sqrt{\tau_\ell}Z$:

$$f_\ell^* = \mathbb{E}[X \mid \mu_\ell^* X + \sqrt{\tau_\ell^*}Z = \cdot], \tag{26}$$

where $\mu_\ell^*$ and $\tau_\ell^*$ are obtained by substituting $f_\ell^*$ as the denoiser in Eqs. (24) and (25).

As with compressed sensing, AMP has powerful theoretical guarantees for rank-one matrix estimation. In some cases, e.g. when $p$ is the uniform distribution over $\{1, -1\}$, the Bayes AMP algorithm is information-theoretically optimal, i.e., it asymptotically matches the MSE achieved by the Bayes optimal estimator [DM14]. In general, Bayes AMP is conjectured to achieve asymptotically optimal MSE over all polynomial-time algorithms for this task [MV21] while being provably optimal over all GFOMs [MW22b].

Despite these guarantees, Bayes AMP for rank-one matrix estimation suffers from the same implementation bottlenecks regarding knowledge of the true prior of the underlying signal. And as with compressed sensing, choosing the state evolution parameters $\mu_\ell$ and $\tau_\ell$ according to the true recursion can cause Bayes AMP to diverge, so in practice one estimates these parameters using the previous iterates. In particular, practitioners typically replace $\tau_\ell$ with $\frac{1}{d}\|x_\ell\|_2^2$ using Eq. (25) and $\mu_\ell$ with $\sqrt{|\frac{1}{d}\|v_\ell\|_2^2 - \frac{1}{d}\|x_\ell\|_2^2|}$ using the infinite dimensional distribution of the components of $v_\ell$. The latter holds when the prior $p$ has zero mean and unit variance; in general, the estimate must be scaled down by an additional factor of $\sqrt{\mathbb{E}_{X \sim p}[X^2]}$, which can be estimated from data. Unless expressed otherwise, we assume $\mathbb{E}_{x \sim p}[x] = 0$ and $\mathbb{E}_{x \sim p}[x^2] = 1$ in this setting throughout. These conditions are without loss of generality for compressed sensing, to which our theoretical results pertain, but not without loss of generality for rank-one matrix estimation.

## C.2 Unrolled architecture

Here, we are given training data $\{(Y^i, x^i)\}_{i=1}^N$ generated by Eq. (21) with all $x^i \sim p_{\mathsf{x}}$. Let $\mathcal{F}$ denote a family of MLPs with fixed architecture constrained to three-dimensional inputs and a one-dimensional output. Set $x_0 = \widehat{1} \in \mathbb{R}^d$ and $v_0 = Yx_0$. Initializing MLP $\widehat{f}_\ell \in \mathcal{F}$ for all $\ell \in [0, L-1]$, we obtain an $L$-layer unrolled network that computes

$$\widehat{x}_{\ell+1} = \widehat{f}_\ell(\widehat{v}_\ell; \widehat{\mu}_\ell, \widehat{\tau}_\ell) \tag{27}$$

$$\widehat{v}_{\ell+1} = Y\widehat{x}_{\ell+1} - \widehat{x}_\ell\langle \partial_1 \widehat{f}_\ell(\widehat{v}_\ell; \widehat{\mu}_\ell, \widehat{\tau}_\ell)\rangle, \tag{28}$$

at every iteration, where $\widehat{\mu}_\ell = \sqrt{|\frac{1}{d}\|v_\ell\|_2^2 - \frac{1}{d}\|x_\ell\|_2^2|}$ and $\widehat{\tau}_t = \frac{1}{d}\|\widehat{x}_\ell\|_2^2$. Here $\widehat{f}_\ell(\,\cdot\,; \widehat{\mu}_\ell, \widehat{\tau}_\ell)$ denotes applying the scalar function $\widehat{f}_\ell(\,\cdot\,; \widehat{\mu}_\ell, \widehat{\tau}_\ell)$ entrywise.

## C.3 Results

**Implementation details.** We set $d = 2^{10}$ and fix $\lambda = 1.5$. We focus on the *Gaussian* and $\mathbb{Z}_2$ priors for this setting. The family of MLPs $\mathcal{F}$ is constrained to have three hidden layers, each with 20 neurons and GELU activations, and we train over a dataset $\{Y^i, x^i\}_{i=1}^N$ of size $N = 2^{12}$ obtained by sampling from the prior and using Eq. (21).

As with compressed sensing, we train with finetuning and also consider a baseline with the MLP denoisers in LDNet replaced with guided denoisers that learn parameters attached to the analytic forms of the Bayes-optimal denoisers.

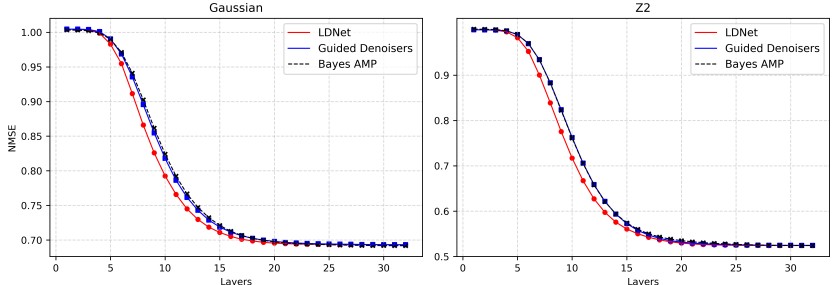

Figure 3: **LDNet for Rank-One Matrix Estimation**. On the left, we plot the NMSE obtained by LDNet and Bayes AMP on the Gaussian prior, while the right plots are on $\mathbb{Z}_2$. LDNet matches Bayes AMP with a slightly quicker convergence.

**Gaussian prior.** For a Gaussian prior, each component $x_i$ of $x$ is drawn $x_i \sim \mathsf{N}(0, 1)$, so $p_x = p^{\otimes d}$ for $p = \mathsf{N}(0, 1)$. As expected, LDNet tracks Bayes AMP to convergence at an NMSE of **0.6931** with a slightly quicker convergence.

$\mathbb{Z}_2$ **prior.** As with compressed sensing, each component of $x$ is drawn from $\{-1, 1\}$ with probability $\frac{1}{2}$. Again, LDNet slightly outperforms Bayes AMP until convergence at an (information-theoretically optimal [DM14]) NMSE of **0.5243**.

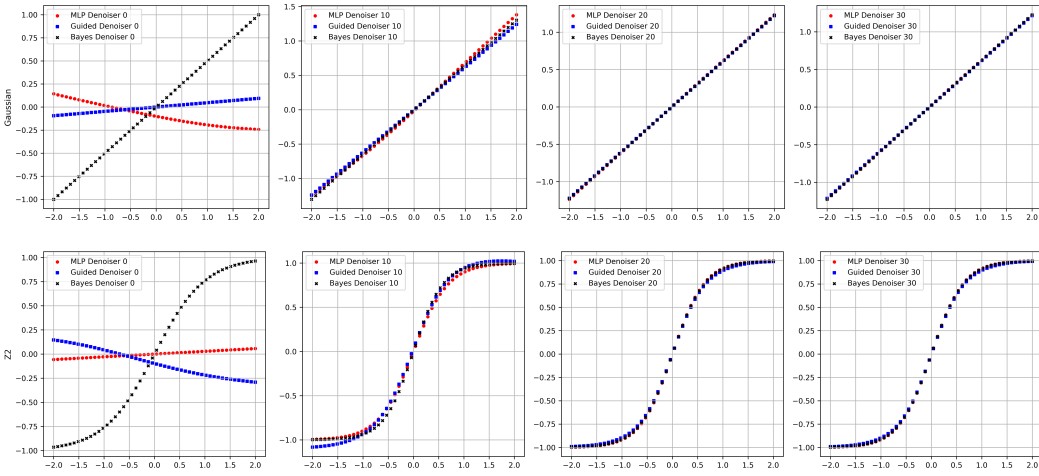

Figure 4: **Learned Denoisers for Rank-One Matrix Estimation**. We plot layerwise denoising functions learned by LDNet on the Gaussian and $\mathbb{Z}_2$ priors relative to their optimal denoisers over a range of inputs in $(-2, 2)$. The state evolution inputs $\mu_\ell, \tau_\ell$ to each denoiser are set to be their empirical estimates.

**LDNet denoisers.** While later iterations are able to recover the Bayes-optimal denoisers, it is worth noting the high approximation error at early iterations as shown in Figure 4. Early iterations

correspond to when the AMP estimates stagnate at an NMSE of roughly 1.0, corresponding to a random, uninformed signal that can accommodate high denoiser error. Around iteration 10 is an inflection point when both LDNet and AMP transition from the uninformed regime to convergence at an informed NMSE (Figure 3), where low approximation error is tolerated.

## D   Beyond Bayes AMP performance

### D.1   Learning auxiliary parameters

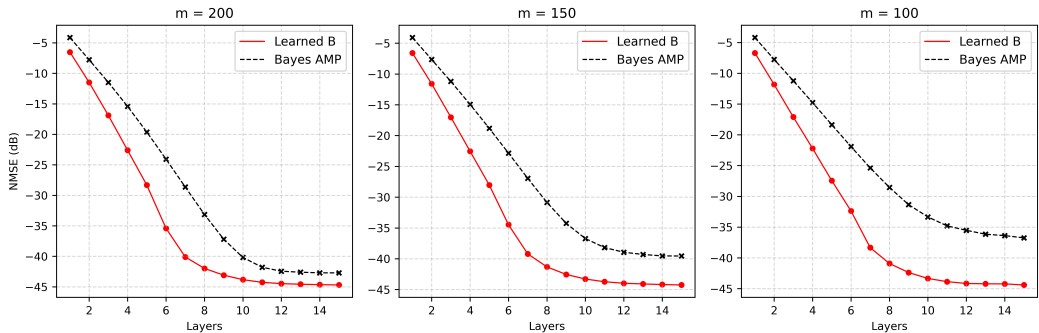

Figure 5: **Learned B with Decreasing Dimension**. We hold $\delta = \frac{1}{2}$ fixed while scaling $m$ from 200 down to 100. Plots show NMSE (dB) performance of unrolling denoisers and learning B vs. Bayes AMP for randomly drawn measurement matrices. There is an increasing gap in performance as $m$ decreases.

In Figure 5, we set the parameters of $B = A^\top$ to be learnable alongside the layerwise denoisers (see Algorithm 2). Holding $\delta = \frac{1}{2}$ fixed while scaling down $m$ has the effect of widening the gap between the NMSE performance of Bayes AMP versus LDNet with trainable $B$. Over five randomly drawn measurement matrices per dimension regime, we find that, on average, LDNet outperforms Bayes AMP by 7.2750% when $m = 200$, 16.4364% when $m = 150$, and 37.0605% when $m = 100$. Here, percentage is measured by $\frac{|\text{Bayes} - \text{LDNet}|}{|\text{Bayes}|} \times 100\%$ in NMSE (dB).

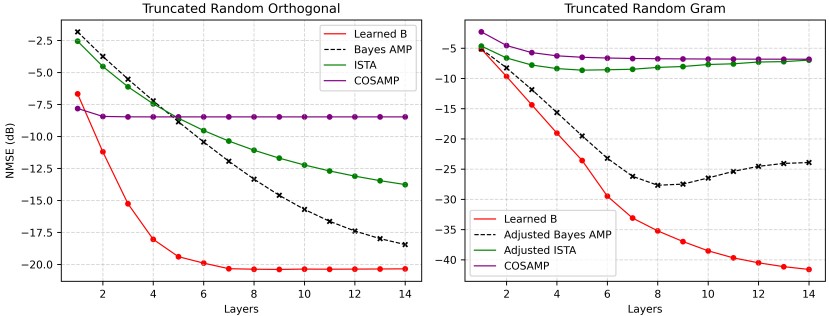

Figure 6: **Non-Gaussian Measurements**. On the left, we plot LDNet with learnable $B$ compared to several baselines for a random truncated orthogonal measurement matrix, and on the right, for a random truncated Gram matrix. LDNet outperforms the other baselines in NMSE as well as convergence.

Another regime to which existing theory for Bayes AMP largely breaks down is when the entries of the sensing matrix $A$ are non-Gaussian. While it was previously observed that learning $B$ can help for ill-conditioned $A$ [BSR17], we find that there are advantages even for well-conditioned (but non-Gaussian) sensing matrices. Figure 6 plots NMSE for two sensing matrices $\tilde{A} \in \mathbb{R}^{250 \times 500}$: one obtained by truncating a random orthogonal matrix $Q \in \mathbb{R}^{500 \times 500}$ (condition number 1), and the other by truncating a Gram matrix $X^\top X \in \mathbb{R}^{500 \times 500}$ with $X_{ij} \sim \mathsf{N}(0, 1/m)$ (condition number 1091). Also displayed are the iterative baselines of Bayes AMP, ISTA, and COSAMP [NT10]. For the truncated random Gram matrix, Bayes AMP and ISTA actually *diverge*, so we plot "adjusted" baselines replacing $\tilde{A}^\top$ in Eqs. (2) and (3) with the "$B$" matrix learned by LDNet. The baselines

---

**Algorithm 2:** Learning $B$

---
**Input:** Training data $\mathcal{D}$, LDNet $\Psi$, Measurement matrix $A$
1   Initialize $B = A^\top$;
2   **for** $\ell = 0$ *to* $\ell = L - 1$ **do**
3     **if** $\ell > 0$ **then**
4        Initialize $\widehat{f_\ell} \leftarrow \widehat{f}_{\ell-1}$;
5     Freeze learnable weights in $B$ and $\widehat{f_k}$ for $k < \ell$;
6     Train $\Psi[0 : \ell]$ on $\mathcal{D}$;
7     Unfreeze learnable weights in $B$ and $\widehat{f_k}$ for $k < \ell$;
8     Train $\Psi[0 : \ell]$ on $\mathcal{D}$;
**Output:** Fully trained $\Psi$, Learned $B$

---

considered all drastically underperform our unrolled network. In fact, the underperformance of "adjusted" Bayes AMP demonstrates that the LDNet denoisers are strongly coupled with $B$, suggesting that learning denoisers is beneficially composable with the traditional algorithm unrolling approach.

All told, we see how adding auxiliary learnable parameters can mitigate scenarios (e.g. finite dimensionality, non-Gaussian or ill-conditioned sensing matrices) where Bayes AMP is suboptimal or not known to be optimal.

### D.2    Non-product priors

Thus far, our theoretical and experimental results have remained in the regime of product priors. But what happens when our underlying signal is drawn from a non-product distribution?

The modifications to AMP are minimal, as detailed in [BMN20], amounting to $d$-dimensional denoisers $f_\ell : \mathbb{R}^d \to \mathbb{R}^d$ and replacing the average derivative of the scalar denoiser in the Onsager term with the normalized divergence $\frac{1}{d}\mathrm{div} f_\ell$ in Eqs. (3) and (23). In this *non-separable* setting, AMP still satisfies a one-dimensional state evolution recursion [BMN20]. In fact, in the asymptotic limit, in some sense our theoretical guarantees carry over if for *generic $d$-dimensional priors*, minimizing the score-matching objective via gradient descent (now in $d$ dimensions instead of 1 dimension) can learn the Bayes-optimal denoiser with gradient descent. This is a question of immense interest within the theory and practice of diffusion generative modeling and remains an important open direction in this area.

In practice, for compressed sensing, unrolled AMP has been shown to be performant on image datasets [MMB17], which serve as prime examples of real-world, non-product signal priors. For our purposes, we focus on rank-one matrix estimation which, even in the product setting, remains unexplored in the unrolling literature. Additionally, we work with handcrafted priors, where we can plot a baseline achieved by Bayes AMP.

**LDNet for non-product priors.** We work in the low-dimensional regime $d = 10$, where $x \in \mathbb{R}^d$. LDNet requires small modifications to the layer iterations defined by Eqs. (27) and (28). The family of MLPs $\mathcal{F}$ parametrizing the denoisers have three hidden layers with 1000 neurons and GELU activations, with input dimension $d + 2$ and output dimension $d$. We take $\widehat{f_\ell} \in \mathcal{F}$ for $\ell \in [0, L - 1]$, and we replace $\langle \partial_1 \widehat{f_\ell} \rangle$ in Eq. (28) with $\frac{1}{d}\mathrm{div}\widehat{f_\ell}$. To avoid backpropagating through the Jacobian during training, we omit the finetuning step in Algorithm 1.

**Signal distributions.** To analyze the performance of $d$-dimensional LDNet, we consider two priors on $x$: product $\mathbb{Z}_2$ and a *mixture of Gaussians*. In both instances we work with a dataset of size $N = 2^{12}$ generated by sampling a signal and using Eq. (21). The product $\mathbb{Z}_2$ prior serves as a test example to see whether the multi-dimensional learnable denoiser provides additional performance gain over Bayes AMP when treating the product distribution as $d$-dimensional as opposed to one-dimensional.

Mixture of Gaussians provide a quite general class of non-product priors. We consider $p = \frac{1}{2}N(\mu_1, \Sigma_1; x) + \frac{1}{2}N(\mu_2, \Sigma_2; x)$, where we choose $\mu_1$ and $\mu_2$ at random with each coordinate chosen from $N(0, 1)$, ensuring that $\mu_1$ and $\mu_2$ do not define the same direction. For each of the covariance matrices, we begin by a choosing random vector with each coordinate drawn uniformly from $[1, 2]$ and normalize so that vector has norm $\sqrt{d}$. We take this to be the diagonalization (i.e. eigenvalues) of the covariance, and conjugate by a randomly drawn orthogonal matrix.

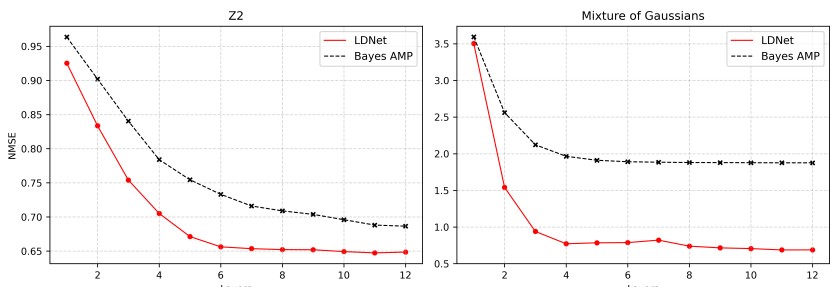

Figure 7: **Multi-Dimensional LDNet for Rank-One Matrix Estimation**. On the left, we plot the NMSE obtained by LDNet and Bayes AMP on $\mathbb{Z}_2$, while the right plots are on the mixture of Gaussians. LDNet outperforms Bayes AMP by significant margins.

As Figure 7 demonstrates, multi-dimensional LDNet significantly outperforms the Bayes AMP baseline on both priors. On a product $\mathbb{Z}_2$ prior, LDNet achieves an NMSE of 0.6485 compared to Bayes AMP's 0.6864, marking a 5.52% improvement while also reaching convergence much faster. On the mixture of Gaussians prior, LDNet achieves NMSE 0.6881 compared to Bayes AMP's 1.8757, marking a 63.32% improvement.

### D.3 Learned denoiser dependence on sensing matrix

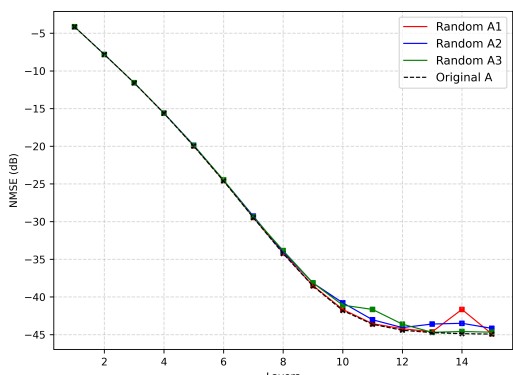

Figure 8: **Transfer Experiments** . Above we plot the NMSE (in dB) over 15 iterations for different choices of measurement matrices coupled with our learned MLP denoisers, including the training-time sensing matrix. We see that the denoising functions are roughly transferable to several random Gaussian measurement settings, suggesting the learning process is not coupled to the fixed sensing matrix seen during training.

Recall that our unrolled denoising network differs from the theoretical setting in two key ways: **a)**, the network is trained assuming a fixed sensing matrix $A$ rather than in expectation over random Gaussian $A$, and **b)**, state evolution parameters are estimated from previous iterates to account for finite dimension corrections.

Indeed, despite these differences, the plots in Figure 2 suggest our network appears to learn a fundamental "optimal" denoiser that is independent of $A$. To further verify this claim, we froze the learned MLP denoiser weights for the Bernoulli-Gaussian prior and replaced the $A$ matrix in Eqs. (5) with other randomly sampled Gaussian matrices $A' \in \mathbb{R}^{250 \times 500}$. As shown in Figure 8, this leads to minimal changes in the NMSE profile.

# E Explicit expressions for various Bayes-optimal denoisers

For prior $p$ and $X \sim p$, $Z \sim \mathsf{N}(0,1)$, the Bayes optimal denoiser in Bayes AMP is given by

$$f_\ell^* = \mathbb{E}[X|X + \tau_\ell Z = y] \tag{29}$$

for compressed sensing and

$$f_\ell^* = \mathbb{E}[X|\mu_\ell X + \sqrt{\tau_\ell} Z = y] \tag{30}$$

for rank-one matrix estimation. For the priors examined in our experiments, we write out the setting-specific optimal denoiser along with the parameterized guided denoiser form (if relevant), where the parameters are learnable during training.

**Bernoulli-Gaussian prior.** One can compute the optimal denoiser to be

$$f_\ell^*(y) = \frac{y}{\left(1 + \tau_\ell^2\right)\left(1 + \frac{1-\varepsilon}{\varepsilon} \frac{\mathsf{N}(0,\tau_\ell^2;y)}{\mathsf{N}(0,\tau_\ell^2+1;y)}\right)}. \tag{31}$$

and parameterize the denoiser via

$$\widehat{f}_\ell(y; \theta_1, \theta_2) = \frac{y}{\left(1 + \frac{\tau_\ell^2}{\theta_1}\right)\left(1 + \sqrt{1 + \frac{\theta_1}{\tau_\ell^2}} \exp\left(\theta_2 - \frac{y^2}{2(\tau_\ell^2 + \tau_\ell^4/\theta_1)}\right)\right)}, \tag{32}$$

as done in [BS16, BSR17].

$\mathbb{Z}_2$ **prior.** The compressed sensing optimal denoiser [DM14] can be written as

$$f_\ell^*(y) = \tanh\left(y \cdot \frac{1}{\tau_\ell^2}\right), \tag{33}$$

which we parameterize as

$$\widehat{f}_\ell(y; \beta) = \tanh\left(y \cdot \beta \frac{1}{\tau_\ell^2}\right). \tag{34}$$

For rank-one matrix estimation, the optimal denoiser can be similarly computed to be

$$f_\ell^*(y) = \tanh\left(y \cdot \frac{\mu_\ell}{\tau_\ell}\right), \tag{35}$$

parametrized as

$$\widehat{f}_\ell(y; \beta) = \tanh\left(y \cdot \frac{\beta \mu_\ell}{\tau_\ell}\right). \tag{36}$$

**Gaussian prior.** For rank-one matrix estimation, the optimal denoiser for a Gaussian prior is

$$f_\ell^*(y) = y \cdot \frac{\mu_\ell}{\mu_\ell^2 + \tau_\ell} \tag{37}$$

parametrized as

$$\widehat{f}_\ell(y; \beta) = y \cdot \beta \frac{\mu_\ell}{\mu_\ell^2 + \tau_\ell}. \tag{38}$$

**Mixture of Gaussians prior.** The calculation of the Bayes-optimal denoiser for a mixture of Gaussians prior in rank-one matrix estimation is slightly more involved, so we provide some more details about the calculation. Given $p = \frac{1}{k} \sum_{i=1}^k \mathsf{N}(\mu_i, \Sigma_i; x)$, where $\mu_i \in \mathbb{R}^d$ and $\Sigma_i \in \mathbb{R}^{d \times d}$ are invertible positive semidefinite symmetric covariance matrices, convolution with $\sqrt{\tau_\ell} Z \sim$

$N(0, \tau_\ell I_d)$ results in the mixture of Gaussians distribution $\tilde{p} = \frac{1}{k} \sum_{i=1}^{k} N(\mu_\ell \mu_i, \mu_\ell^2 \Sigma_i + \tau_t I_d; x)$. For any mixture $\sum_{i=1}^{k} \lambda_i N(\mu_i, Q_i; x)$, the score is given by [CKS24]

$$- \sum_{i=1}^{k} \left( \frac{\lambda_i N(\mu_i, Q_i; x)}{\sum_j \lambda_j N(\mu_j, Q_j; x)} \right) Q_i^{-1}(x - \mu_i). \tag{39}$$

Thus, we have

$$\nabla \log \tilde{p}(x) = - \sum_{i=1}^{k} \frac{N(\mu_\ell \mu_i, \mu_\ell^2 \Sigma_i + \tau_t I_d; x)}{\sum_j N(\mu_\ell \mu_j, \mu_\ell^2 \Sigma_j + \tau_t I_d; x)} (\mu_\ell^2 \Sigma_i + \tau_t I_d)^{-1}(x - \mu_\ell \mu_i), \tag{40}$$

so by Tweedie's formula, the posterior mean on $X$ given $\mu_\ell X + \sqrt{\tau_\ell} Z = y$ is

$$\frac{1}{\mu_\ell} \cdot \left( y - \left( \tau_\ell \cdot \sum_{i=1}^{k} \frac{N(\mu_\ell \mu_i, \mu_\ell^2 \Sigma_i + \tau_t I_d; y)}{\sum_j N(\mu_\ell \mu_j, \mu_\ell^2 \Sigma_j + \tau_t I_d; y)} (\mu_\ell^2 \Sigma_i + \tau_t I_d)^{-1}(y - \mu_\ell \mu_i) \right) \right). \tag{41}$$

