# OpenReview forum: "Unrolled denoising networks provably learn to perform optimal Bayesian inference"
_NeurIPS.cc/2024/Conference — NeurIPS 2024 poster_

### Official Review · Reviewer_mdyi · 2024-07-12

**Soundness:** 3
**Presentation:** 3
**Contribution:** 3
**Rating:** 7
**Confidence:** 4

**Summary:**

This work seeks to understand why neural network-based approaches for inverse problems can outperform methods that incorporate hand-crafted priors. In the Bayesian setting, it is known that when the true prior is available, the optimal estimator (in a mean square sense) is the conditional mean. However, in practice, the true prior is never known, and learning-based methods, such as those based on unrolling, must implicitly estimate such a prior from samples of the unknown prior. While such approaches have shown to perform well empirically, it is unclear from a mathematical perspective if they are learning the true prior. This work addresses this by analyzing what estimators do unrolled AMP-based networks converge to. It is shown that when the underlying prior comes from a product distribution with subGaussian coordinates, then the network trained with SGD will converge to the optimal denoiser, if the prior were known. This convergence occurs in the high-dimensional asymptotic regime. The authors support their theoretical results with empirics showing that the learned denoisers converge to the optimal denoisers under the setting of their theorem, along with extensions to other problems, such as the nonlinear rank-1 matrix estimation problem.

**Strengths:**

- The work is tackling an interesting and timely question, which is why do neural-network based methods outperform algorithms based on hand-crafted priors and can they reach the performance of “optimal” methods.
- Toy empirical results help in supporting the theory that the training scheme and network architectures converge to the optimal Bayes denoisers.

**Weaknesses:**

- Some aspects regarding the readability and presentation could be improved, especially as they relate to the main theorems. I have noted my questions below.
- The assumptions on the prior distribution being a smooth product distribution feels somewhat limiting. In particular, the authors note that their overparameterization result is dimension-free, but I wonder whether this is a result of the fact that with a smooth product distribution, the dynamics are characterized by the 1-d state evolution. Do the authors have a sense of whether the result would still be dimension-free if the product distribution assumption was relaxed?

**Questions:**

- I am confused about how the notion of complexity is applied in the Theorem 2. In particular, what is the function $\phi$ in the statement of Theorem 2? Is it the underlying “optimal” denoiser $f_{\ell}^*$ or is it the one-hidden layer ReLU network? It is also not clear in the statements of Lemma 3 and Theorem 3 in the appendix what the function classes $\mathcal{F}$ are. Are these functions of the form $F^*$ with bounded coefficients $(a_i, w_{i,1},w_{i,2})$? Moreover, to apply Lemma 3, the $\phi_i$’s need to be infinite-order smooth. How does this arise in the assumptions of the Theorem?
- Also, why is the error on the right-hand side of the inequality in terms of $\hat{v}_L$ and $\hat{x}_L$ instead of $v_L$ and $x_L$, the outputs of the optimal Bayes AMP? It’s not clear why it should be in terms of the outputs of the learned AMP, since the claim is that the learned AMP’s ell2 error is close to the optimal Bayes AMP ell2 error.
- Have the authors conducted experiments in the case when the distribution is not a product prior? Do similar results extend to this case?

**Limitations:**

The authors discuss the limitations of their theory in the Conclusion section, noting that the assumptions on the prior distribution and linearity/Gaussianity of the inverse problem are in particular, limited settings. There are no broader societal impacts.

---

> ### Author Rebuttal · Authors · 2024-08-07
>
> We thank the reviewer for their thoughtful questions and remarks.
>
> - **On the smoothness and product distribution assumption**: Please see the global response for the discussion about the assumptions on the prior.
>
> - **On the notion of the complexity of denoisers**:  We think that this confusion is because of a typo/underspecification in the statement of theorem 2 and theorem 3. The notion of complexity is applied to the underlying optimal denoiser $f_\ell^*$ (i.e. the function $\phi$). In theorem 2/theorem 3, $M_0$ and $N_0$ are polynomial in the sum of the complexities of $f_{\ell}^*$ for $\ell$ from $0$ to $L-1$. Moreover, $M_0$ and $N_0$ are polynomial in $\sum_{\ell \in [0, L-1]} C(f_{\ell}^*, R(\log \frac{1}{\epsilon})^{1.5})$ and $\frac{1}{\epsilon}$. We apologize for the confusion caused by the typo and will update it in the revised draft of the paper.
>
> - **On applying function classes and Lemma 3**:
>    - When we refer to the function class $\mathcal{F}$, we are indeed referring to target functions $F^*(x)$ with bounded coefficients. Moreover, to obtain the learning result for the $\ell^{th}$ layer denoiser, we apply Lemma 3 representing $f_{\ell}^*(x)$ as a target function $F^*(x)$.
>    - Note that this is equivalent to setting $p=1$ and $a_1=1$, along with $\phi_1 = f_\ell^*$ (where $a_1^*=1$ and $w_{1, 1}^*=\langle 1, 0\rangle $, $w_{1, 2}^*=\langle 0, 1\rangle$; note we are treating "$x$" as the two dimensional input $\langle x, 1 \rangle$ into the target function).
>    - Recall that $f_{\ell}^*$ is the optimal denoiser at noise level $\tau_\ell^*$ (i.e., $f_{\ell}^*(y) = E[ \mathbf{x} | \mathbf{x} + \tau_\ell^* \mathbf{z} = y ]$); therefore, it is an infinite order smooth function. To get an intuition for why this is the case, note that by Tweedie’s formula, the optimal denoiser is, up to a shift by $x$ and scaling, given by the derivative of $p(x; \tau_\ell^*)$, i.e. the log-density of the prior convolved with some Gaussian. Even if the prior were a mixture of Dirac deltas, the prior convolved with Gaussian is a mixture of Gaussians, and the log-density of this has derivatives at all orders. In particular, $p(x; \tau_\ell^*)$ is always positive and continuous (due to the convolution with a Gaussian, no matter the prior), so all higher order derivatives of $\frac{\nabla p(x; \tau_\ell^*)}{p(x; \tau_\ell^*)}$ exist by applying the quotient rule.
>    - We thank the reviewer for bringing up these details and will expand on them in the revisions.
>
> - **On the right-hand-side error of the inequality**: This is a typo. The right-hand side of theorem 2 should be $v^L$ and $x^L$ instead of $\hat{v}^L$ and $\hat{x}^L$ as mentioned in the formal version of the theorem in the appendix (theorem 3). This follows from the closeness of the state-evolution parameters ($\tau^L$ and $\hat{\tau}^L$) of iterates $x^L$ and $\hat{x}^L$. We thank the reviewer for pointing this out, and we will update it in the revised draft.
>
> - **On experiments with non-product signal priors**: We have subsequently extended our experiments to non-product priors. We do see similar advantages to unrolling over Bayes-AMP show up in this setting; please see the global response and the supplemental second figure in the attached pdf for a discussion of the non-product prior experiments.

---

> > ### Comment · Reviewer_mdyi · 2024-08-08
> > **Response to authors**
> >
> > Thank you to the authors for their detailed response and answering my questions. I appreciate the additional experiments that were conducted on examples with non-product distributions. I was hoping to further ask about the sample complexity:
> > - **Notion of complexity:** Thank you for clarifying that the function $\phi$ indeed should be the optimal denoiser $f_{\ell}^*$. I was hoping to make sure I understand this point further. If the notion of complexity then depends on $f_{\ell}^*$, how should I think about how the quantity $C(f_{\ell}^*, R(\log1/\epsilon)^{1.5})$ scales depending on the distribution $p$ chosen? In particular, you mention in another bullet point that $f_{\ell}^*$ is an infinite-order smooth function under the Theorem's assumptions. In this case, what would the complexity quantity scale like? There is an example in the paper stating that polynomials of degree $\ell$ have complexity $\mathrm{poly}(\alpha^{\ell},\log(1/\epsilon)^{\ell})$ and that for cases where the score is sufficiently smooth, one could apply Jackson's theorem to approximate them. What will be the tradeoff in terms of approximation in sample complexity (e.g., the approximation error + complexity when choosing an $n$-degree polynomial)? Are there examples of distributions $p$ for which we can explicitly calculate or easily bound $C(f_{\ell}^*, R(\log1/\epsilon)^{1.5})$?

---

> > > ### Author Response · Authors · 2024-08-08
> > >
> > > We thank the reviewer again for clarifying the subtle details on how the complexity quantity scales with the denoiser.
> > >
> > > In Definition 1 in the submission, the complexity $\mathcal{C}(\phi, \alpha)$ is given by the equation
> > >
> > > $\mathcal{C} ( \phi, \alpha ) = \sum_{i=0}^\infty \Big( \frac{ \alpha C \sqrt{ \log (1 / \epsilon) } }{ \sqrt{i} } \Big)^i | c_i |$.
> > >
> > > Since we know that the denoisers $\phi = f_\ell^*$ are infinite-order smooth, we can leverage Jackson’s Theorem to get a polynomial approximation of $f_\ell^*$ up to degree $k$, and this degree $k$ is what controls how $\mathcal{C}(\phi, \alpha)$ scales.
> > >
> > > When $\alpha = R(\log 1/\epsilon)^{1.5}$, plugging into definition 1, the leading term takes the form $\frac{\sqrt{k} {R^{k}}}{k^{\frac{k}{2}}} \cdot (\log{1/\epsilon})^{2k} || c ||$ where $||  c ||$ denotes $L_2$ norm of $[c_0,  c_1, \ldots, c_k]$ vector.
> > >
> > > As you pointed out, the key tradeoff is the approximation error in Jackson’s theorem, which modulates the degree of the polynomial expansion, which dictates how this leading term scales.
> > >
> > > Jackson’s theorem states that the error of approximation $f_\ell^*$ with a degree $k$ polynomial is at most a constant times $|\nabla f_\ell^*|/k$. Since we further assume the score function is $B$-Lipschitz, this is at most $B/k$. Thus, if we want an approximation error $\delta$ in the polynomial approximation, we require $k \geq \frac{B}{\delta}$.
> > >
> > > Putting this all together, if you admit error $\delta$ in the polynomial approximation of $f_\ell^*$ and error $\epsilon$ in SGD, then the complexity scales polynomially with leading term
> > >
> > > $\left(\frac{\delta}{B}\right)^{\frac{B}{2\delta}} \cdot (R(\log{1/\epsilon})^{2})^{B/\delta} || c ||$.
> > >
> > > **Example**: For $Z_2$ prior, we know that the denoiser function is given by $\tanh(y / \tau_\ell^2)$. In this case, the denoiser function is Lipschitz function with $\frac{1}{\tau_\ell^2} \leq \frac{1}{\sigma^2}$ constant. Additionally, it is sub-Gaussian with constant sub-Gaussianity constant. Moreover, $|| c || \leq O((R \log 1/\epsilon)^{10B/\delta})$ (see Lemma 23 in [1], the polynomial in Lemma 23 is bounded with constant at most 2 for $\delta < 1$ because it is $\delta$-approximation of $\tanh$ (a bounded function)). Using this bound on $|| c ||$ and $B=1/\sigma^2, R=O(1)$, the complexity of $\delta-$approximation of the denoiser is
> > >
> > > $\left( \delta \sigma^2 \right)^{\frac{1}{2\sigma^2 \delta}} \cdot (C \log{1/\epsilon})^{ \frac{12}{\sigma^2\delta} }$
> > >
> > > for some large constant $C$.
> > >
> > >
> > > We will include this explanation in the appendix of our revision for clarity. Thank you again for engaging closely with our work!
> > >
> > >
> > > [1] Goel et al. Reliably Learning the ReLU in Polynomial Time.

---

> > > > ### Comment · Reviewer_mdyi · 2024-08-09
> > > >
> > > > Thank you to the authors for this detailed discussion on the sample complexity. This really helps in terms of better understanding how to apply these bounds and making them more transparent. It's a good idea to include them in the appendix as illustrative examples of the theory.
> > > >
> > > > Based on our discussion, I believe my comments about readability have been addressed. While the smooth product prior distribution still feels like a limitation, the experiments suggest that there are interesting avenues for future work. As such, I will raise my score.

---

### Official Review · Reviewer_DSkx · 2024-07-12

**Soundness:** 3
**Presentation:** 4
**Contribution:** 3
**Rating:** 6
**Confidence:** 3

**Summary:**

This work investigates unrolling approximate message passing (AMP) for solving compressive sensing under Gaussian design and separable prior distribution. The unrolling scheme consists of parametrizing the AMP denoiser at each iteration by the layer of a neural network, which is sequentially trained using observations from the signal distribution by minimizing the empirical mean-squared error.

The main result is to show that with a polynomially wide two-layer neural network trained under one-pass SGD with with polynomially many observations, unrolled AMP asymptotically achieves the same MSE as AMP implemented with the optimal denoiser.

**Strengths:**

The paper is well-written and easy to follow. It is also self-contained - the small introduction to AMP in Section 2.1 is a nice addition since most NeurIPS readers will not be familiar with AMP. The results are interesting, since they suggest that the optimal AMP denoiser can be learned "on the fly" with unrolling.

**Weaknesses:**

The topic is relatively niche within the NeurIPS community. Also, AMP is designed to Gaussian measurements, and it is well-known that as an algorithm it is not very robust to other designs, therefore the results have a limited practical scope. Unrolling requires samples from the signal distribution.

**Questions:**

- L80-81:
> "*and thus achieve mean-squared error which is conjectured to be optimal among all polynomial-time algorithms*"

As the authors probably know, there are a few counter examples to this conjecture, including for problems which are closely related to Compressive Sensing (e.g. noiseless phase retrieval, see []). I suggest the authors being more precise, by either refeering to the known optimality results within first order methods [CMW20], MW22b] or by adding a footnote saying the conjecture is believed to hold in a broad sense, since all known exceptions rely on rather fine tuned algorithms (like LLL in the noiseless phase retrieval case)

- The observation in L303-304 that the denoiser learned by unrolling can outperform the optimal denoiser is surprising. I understand that optimality of AMP is an asymptotic statement - so this difference should get smaller with the dimension - but is this something you consistently observed throughout in the experiments? Is this improvement always within a $O(d^{-1/2})$ interval? If yes, how can it be distinguished from finite-size fluctuations?

- Can the authors comments what "*sufficiently small*" learning rate and "*sufficiently large*" number of steps exactly mean?

- The tricks for training discussed in Appendix B are intriguing. Do the authors have an intuitive understanding of why fine-tuning is necessary for successfully learning the a good denoiser for Compressive sensing with Gauss-Bernouilli prior but not rank-one matrix factorization and Compressive sensing with $\mathbb{Z}_{2}$?

- At a similar note to the above, what is the intuition of why sequential learning avoid bad local minima? Similarly, why initializing at the solution of the previous layer avoids local minima? How these local minima look like and how they compare with the optimal denoisers? This should be easy to visualise since the denoiser is a one dimensional function (as in Figure 2)

- It would help readability to add in all figure captions the details on the plots, e.g. the dimensions $m, n, d, L$.

**Limitations:**

Limitations are discussed in Section 5.

---

> ### Author Rebuttal · Authors · 2024-08-07
>
> We thank the reviewer for their detailed comments.
>
> - **On a relatively niche topic within the NeurIPS community**: While understanding the Bayes-AMP algorithm in itself is a relatively niche area for the NeurIPS community, our theoretical results show that the learnability of Bayes-AMP amounts to characterizing training dynamics for the score-matching objective. Given the prevalence of score matching within the literature on diffusion models, which is of very wide interest in the NeurIPS community, we believe that this line of work involving unrolling and AMP offers valuable insight into the dynamics and approximation capabilities of diffusion models. Some of this has been explored in work by [Mei24] and [MW24] (see our related work section in the Appendix).
>
> [Mei24] Song Mei. U-nets as belief propagation: Efficient classification, denoising, and diffusion in generative hierarchical models.
>
> [MW23] Song Mei and Yuchen Wu. Deep networks as denoising algorithms: Sample-efficient learning of diffusion models in high-dimensional graphical models.
>
> - **AMP on non-Gaussian measurements**:
>   - We agree with the reviewer that AMP has these limitations, but empirically we find that *unrolling offers a powerful way to interpolate between the theoretically optimal guarantees of AMP in “nice” Gaussian settings and the practically useful guarantees of neural network-based methods for inference in real-world settings.*
>   - For example, we find in our “learnable B” experiments that by learning auxiliary matrix parameters, unrolling can surpass the performance of Bayes AMP (and baselines such as ISTA) on non-Gaussian designs despite working with an architecture inspired by AMP. In particular, we examine a random orthogonal design (fig. 4) and a random Gram matrix design (attached fig. in pdf in global response) and show our method can surpass Bayes-AMP. While our theory only explains why unrolling can compete with AMP, it opens up the potential for future work exploring why learning auxiliary matrix parameters can ameliorate the non-robustness of AMP to other designs.
>
> - **On the conjectured to be optimal among all poly-time algorithms**: We thank the reviewer for bringing this up. We will in our revision refer to the precise GFOM optimality results for compressed sensing and low-rank matrix estimation to qualify the optimality of AMP.
>
> - **On the denoiser learned by unrolling can outperform the optimal denoiser**:
>   - With regards to learning the denoiser, as in Figure 1, the unrolled denoisers outperform Bayes AMP by a relatively small margin. The NMSE in Figure 1 is averaged over $2^{15}$ samples with fixed, randomly chosen design, and we observed this improvement consistently for other randomly sampled designs as well, so we do not believe the small improvement of MLP denoisers over vanilla AMP in Figure 1 is from random fluctuations. The broader point we want to emphasize is that Bayes AMP is only known to be “optimal” up to $o_d(1)$ margin in NMSE, and the fact that the MLP denoisers we learn improve over this in the finite-dimensional setting is consistent with this state of knowledge.
>   - We also observed consistently that as the signal dimension decreased, we could outperform Bayes AMP by learning the “B” matrix (or measurement transpose) as detailed in Figure 3. Here, we believe the learned B experiments take advantage of finite-dimensional sub-optimalities of AMP as a GFOM, rather than just finite-dimensional sub-optimalities of the denoisers used by Bayes AMP.
>
> - **On the intuitive understanding of the fine-tuning**:
>   - We observed finetuning was not largely impactful for the compressed sensing experiments but was helpful for rank-one PCA. Note that PCA has different strengths of initialization for AMP of the signal estimate while compressed sensing has a “canonical” initialization ($x_0=0$). PCA thus induces a “burn-in” phase where the AMP estimates reflect almost no signal at early iterations (see Figure 6 of the submission).
>   - We suspect that finetuning assists in maintaining the consistency of the learned denoisers across this burn-in phase where the correlation between the estimate and MMSE estimate is low, communicating the loss objectives of the later denoisers to earlier iterations.
>   - We also observe that finetuning improves validation loss most during this “burn-in” phase, but notably not as much during later iterations where the estimate is close to the MMSE estimate.
>
> - **On the intuition of why sequential learning avoid local minima**:
>   - Sequential learning takes advantage of the following result: Suppose we have learned the Bayes-AMP denoisers in all previous layers. Then the denoiser that minimizes the MSE estimate of the next layer is the corresponding Bayes-AMP denoiser as well (see thm 3 in [MW22]). This simplifies the learning optimization immensely, as we only need the current layer’s denoiser to globally optimize the MSE loss function.
>   - Another way to view this is that we know that our learned network should provide the minimum MSE estimate after each layer, but training end-to-end throws away this “intermediate layer” information and only uses the final layer’s optimality.
>
> [MW22] Statistically Optimal First Order Algorithms: A Proof via Orthogonalization. Andrea Montanari, Yuchen Wu.
>
> - **Intuition behind initializing at the solution of the previous layer**: As for initializing at the solution of the previous layer, the intuition is as follows. As the Bayes-AMP estimates converge, so do the denoisers, so we expect that the denoiser from the previous layer is close to the optimal denoiser, at least closer than a random initialization.
>
> - **On quantification of learning rate and number of steps**: By sufficiently small learning rate, we mean that the learning rate is $\Theta(1/(\epsilon*m))$ and by a sufficiently large number of steps, we mean polynomial in the complexity of the denoiser and $1/\epsilon$ (See the statement of Theorem 3).

---

> > ### Comment · Reviewer_DSkx · 2024-08-14
> >
> > I thank the authors for their rebuttal, which has addressed my questions. I maintain my score and recommendation towards acceptance.

---

### Official Review · Reviewer_F56a · 2024-07-16

**Soundness:** 4
**Presentation:** 4
**Contribution:** 3
**Rating:** 8
**Confidence:** 4

**Summary:**

This paper studies the theoretical capabilities of unrolled denoising networks in the context of compressed sensing and rank-one PCA problems, with a focus on experimental validation. The authors present the first proof of learning guarantees for neural networks based on unrolling Approximate Message Passing (AMP). They demonstrate that, when trained with gradient-based methods, these networks can achieve performance on par with optimal prior-aware algorithms. Specifically, for compressed sensing under the assumption that the data is sampled from a product prior, the parameters of the network converge to the same denoisers used in Bayes AMP. The technical aspect of this proof employs a combination of state evolution and neural tangent kernel (NTK) analysis. Numerically, the authors validate their claims and show that the unrolled architecture can handle more general priors.

**Strengths:**

The paper presents significant theoretical contributions to an important subfield of machine learning. It is well-motivated, clearly written, and aligns with the existing literature. This is a significant theoretical advancement that bridges the gap between optimal prior-aware algorithms and data-driven training methods.

The authors introduce a novel proof technique that combines state evolution and Neural Tangent Kernel (NTK) analysis. Unlike prior work, this proposed analysis is robust and independent of dimensionality.

The author provides sufficient numerical evidence to validate their theoretical contributions.

**Weaknesses:**

The paper's theoretical results are confined to settings of compressed sensing with data derived from a product prior, which is an unrealistic assumption compared to data from actual science and engineering applications.

Furthermore, the chosen architecture could be considered impractical in practice due to its fixed width and depth.

However, these weaknesses also present opportunities for future research.

**Questions:**

* Have the authors numerically simulated the effects of different network architectures on the robustness of their estimates? This inquiry focuses on understanding how architectural variations impact the results, rather than solely assessing performance.

* Could the authors comment on the practical likelihood of Assumption 1?

* Could the authors please provide context for the variable B in Informal Theorem 2? Its meaning is clear from Formal Theorem 3, but not from the informal statement.

**Limitations:**

Limitations are clearly discussed

---

> ### Author Rebuttal · Authors · 2024-08-07
>
> We sincerely thank the reviewer for their effort in reviewing our work and address their comments below.
>
> - **On the product distribution assumption**: Please see the global response for the discussion about the assumptions on the prior. We have also subsequently expanded experiments to the non-product distribution regime, also included in the global response and the second supplementary figure in the attached pdf.
>
> - **On the flexibility of the architecture of the MLP denoiser**: The MLP parameterizing the denoiser can in practice be set to any architecture (convolutional net, U-net, etc.), and doesn’t have to be constrained to the three-layer, 70-neuron wide MLP we used.
>
> - **On the understanding how architectural variations impact the results**: We examined different depths and widths of our MLP parameterization and found that the behavior of unrolling was relatively robust to variations in the choice of the denoiser architecture. The adjustments we made were ultimately meant to control for the amount of error in learning the denoiser, and the architecture we eventually reported results for worked sufficiently well for our experiments while also being quite reasonable to train.
>
>
> - **On the practical likelihood of Assumption 1**: Assumption 1 comprises two parts: 1) sub-Gaussianity and 2) Lipschitz continuity of the $\tau-$Gaussian smoothed score function (score function of $x + \tau z$). We believe both assumptions are mild, hold for a large distribution class, and are used in prior literature (e.g., the sub-Gaussianity of a distribution holds for any distribution with bounded support and the Lipschitz continuity of the score function holds for mixtures of spherical Gaussians with bounded means). Please also see the general response on the additional discussions about these assumptions. We will include a discussion about the generality of the assumption in the revised draft.
>
>
> - **The definition of variable $B$**: The variable $B$ denotes the Lipschitz constant of the score function of $p(x; \tau)$ where $p(x; \tau)$ is a probability density of random variable $x + \tau z$ where $x$ and $z$ corresponds to the data distribution and standard Gaussian distribution respectively (Please see Assumption 1 for the complete definition of variable $B$).

---

> ### Comment · Reviewer_F56a · 2024-08-11
>
> Thank you for your detail responses I will update my score accordingly to the thoughtful respones the authors have provided.

---

### Author Rebuttal · Authors · 2024-08-07

We thank all the reviewers for their careful reviews and constructive feedback. We address some general comments raised by multiple reviewers here and address the rest of the comments in the individual responses.

- **On the product prior assumption**: We restricted our results to product priors for two main reasons:
   - The product prior setting is quite standard and widely studied within the theory literature on AMP (e.g., see [BM11, DAM15]).
   - The state evolution for AMP, which establishes that the input to the denoisers behaves like samples from the data distribution smoothed with Gaussian noise, still holds for non-product distributions (see [BMN17]). So to show that unrolled AMP provably learns to perform Bayesian inference, one needs to show that unrolled AMP learns denoisers at different noise scales. However, this is a major open problem in the theory of diffusion models (understanding when “score estimation” can be performed in polynomial time) and therefore remains well outside the scope of this work. However, with product distributions, corresponding to scalar/one-dimensional score estimation, we are able to show such a result. That being said, it would be interesting to study the specific class of non-product priors given by **generative priors** coming from pushforwards along random neural networks, as studied by Aubin et al. [ALB20].


[BM11] Mohsen Bayati and Andrea Montanari. The dynamics of message passing on dense graphs, with applications to compressed sensing.

[DAM15] Yash Deshpande, Emmanuel Abbe, Andrea Montanari Asymptotic Mutual Information for the Two-Groups Stochastic Block Model.

[BMN17] Raphaël Berthier, Andrea Montanari, and Phan-Minh Nguyen. State Evolution for Approximate Message Passing with Non-Separable Functions.

[ALB20]  Benjamin Aubin, Bruno Loureiro, Antoine Baker. Exact asymptotics for phase retrieval and compressed sensing with random generative priors.

- **On the smoothness assumption**:
   - We want to reiterate that our theoretical result only requires smoothness of the data distribution after the Gaussian convolution operation. More specifically, we require the Lipschitz constant of the score function of random variable $x + \tau z$, where $x$ is drawn from the prior and $z$ follows the standard Gaussian distribution. Note that this is a mild assumption and has been used in prior literature [CCL+23, CLL23].

[CCL+23] Sampling is as easy as learning the score: theory for diffusion models with minimal data assumptions. Sitan Chen, Sinho Chewi, Jerry Li, Yuanzhi Li, Adil Salim, Anru R. Zhang. 2023.

[CLL23] Improved Analysis of Score-based Generative Modeling: User-Friendly Bounds under Minimal Smoothness Assumptions. Hongrui Chen, Holden Lee, Jianfeng Lu 2023.


- **Extension to experiments with a non-product prior**:
   - We have subsequently extended our empirical results to non-product priors. Here, the learned denoiser is now a $d$-to-$d$ dimensional function as opposed to a scalar function, and the Onsager term is the averaged divergence of the denoiser. Our MLP architecture parametrizes this $d$-to-$d$ input-output structure as well. Our non-product setting is a mixture of Gaussians prior which in general can accommodate practical distributions through kernel density estimation. We outperform the Bayes-AMP baseline in low-dimensional settings; see Figure 2 in the attached pdf.

---

### Decision · Program_Chairs · 2024-09-25

**Decision:**

Accept (poster)

**Comment:**

In recent years, algorithm unrolling has emerged as a deep learning approach to deal with  the origin of priors in bayesian inference: it involves designing a neural network whose layers can, in theory, simulate iterations of these "optimal" inference algorithms, and then train the network on data generated from an unknown prior.  Despite its empirical successes, the theoretical reasons for the effectiveness of this approach and the nature of the estimators these networks converge to have remained unclear. This work claims to provides the first rigorous learning guarantees for neural networks based on unrolling approximate message passing (AMP). In the context of compressed sensing, it is proven that when trained on data generated from a product prior, the network’s layers approximately converge to the same denoisers used in Bayes AMP. The authors also carry out numerical experiments to further demonstrate the advantages of this unrolled architecture: it can adapt to general priors without explicit knowledge, handle more general measurement scenarios (including non-sub-Gaussian measurements), and shows non-asymptotic improvements over the mean squared error achieved by Bayes AMP.

The reviewers all thought the paper was interesting. They raised some technical concerns that were mostly addressed during the response and revision. Given the rating and the discussions this is a clear accept.